# Characterization of a novel interaction of the Nup159 nucleoporin with asymmetrically localized spindle pole body proteins and its link with autophagy

**Inés García de Oya, Javier Manzano-López, Alejandra Álvarez-Llamas, María de la Paz Vázquez-Aroca, Cristina Cepeda-García, Fernando Monje-Casas** *

Centro Andaluz de Biología Molecular y Medicina Regenerativa (CABIMER) / Spanish National Research Council (CSIC)—University of Seville—University Pablo de Olavide, Sevilla, Spain

* fernando.monje@cabimer.es

## Abstract

Both the spindle microtubule-organizing centers and the nuclear pore complexes (NPCs) are convoluted structures where many signaling pathways converge to coordinate key events during cell division. Interestingly, despite their distinct molecular conformation and overall functions, these structures share common components and collaborate in the regulation of essential processes. We have established a new link between microtubule-organizing centers and nuclear pores in budding yeast by unveiling an interaction between the Bfa1/Bub2 complex, a mitotic exit inhibitor that localizes on the spindle pole bodies, and the Nup159 nucleoporin. Bfa1/Bub2 association with Nup159 is reduced in metaphase to not interfere with proper spindle positioning. However, their interaction is stimulated in anaphase and assists the Nup159-dependent autophagy pathway. The asymmetric localization of Bfa1/Bub2 during mitosis raises the possibility that its interaction with Nup159 could differentially promote Nup159-mediated autophagic processes, which might be relevant for the maintenance of the replicative lifespan.

## Introduction

The microtubules that constitute the mitotic spindle, position this structure within the cell, and enable its function in chromosome segregation, emanate from microtubule-organizing centers (MTOCs) located at both spindle poles [1]. The MTOCs, named centrosomes in mammalian cells and spindle pole bodies (SPBs) in the budding yeast *Saccharomyces cerevisiae*, are fundamental players in the regulation of cell division [1]. Besides their essential role in genome distribution, centrosomes and SPBs are platforms where many cell signaling pathways converge to regulate different aspects of mitotic progression [2]. In this way, most constituents of the mitotic exit network (MEN), a signaling cascade that triggers exit from mitosis in *S. cerevisiae*, are localized to the SPBs [3]. This is the case of Bfa1/Bub2, a two-component GTPase-activating protein (GAP) that inhibits MEN signaling and constitutes a central target of the main cell cycle checkpoints [4–6]. Bfa1 and Bub2 integrate signals from multiple sources in order to

**Data Availability Statement:** All relevant data are within the paper and its Supporting information files.

**Funding:** This work was supported by MCIN/AEI/ 10.13039/501100011033/"ERDF A way of making Europe" (grants BFU2013-43718-P and BFU2016-76642-P to F.M.-C.), MCIN/AEI/10.13039/ 501100011033 grant PID2019-105609GB-I00 to F. M.-C.) and MCIN/AEI/10.13039/501100011033/ "ESF Investing in your future" (predoctoral research contracts BES-2017-080805 to A.A.-L. and PRE2020-093933 to M.P.V.-A.). The funders had no role in study design, data collection and analysis, decision to publish, or preparation of the manuscript.

**Competing interests:** The authors have declared that no competing interests exist.

**Abbreviations:** AID, auxine inducible degron; APC, anaphase-promoting complex; BiFC, bimolecular fluorescence complementation; DDC, DNA damage checkpoint; DID, dynein interaction domain; FEAR, Cdc14 early anaphase release; GAP, GTPase-activating protein; GBP, GFP-binding protein; GFP, green fluorescent protein; MEN, mitotic exit network; MTOC, microtubule-organizing center; NPC, nuclear pore complex; SAC, spindle assembly checkpoint; SPB, spindle pole body; SPOC, spindle position checkpoint; TCA, trichloroacetic acid; VC, Venus fluorescent protein C-terminal halve; VN, Venus fluorescent protein N-terminal halve.

coordinate mitotic exit with the successful completion of key cellular events. To this end, the GAP complex is regulated by different kinases that control its activity and/or localization, such as the Polo-like kinase Cdc5, which phosphorylates Bfa1/Bub2 in anaphase to restrain its inhibitory action on the MEN [6]. Additionally, when the spindle position checkpoint (SPOC) is triggered as a consequence of spindle misalignment, Bfa1/Bub2 phosphorylation by the Kin4 kinase prevents the inhibitory action of Cdc5 on Bfa1/Bub2, thereby impeding mitotic exit until the spindle is finally correctly positioned along the mother-daughter cell axis [7,8]. Despite a lot of work has been put into understanding the mechanisms that control the activity and localization of Bfa1/Bub2, many aspects of their regulation are nonetheless still unknown.

*S. cerevisiae* displays a closed mitosis and the SBPs are embedded in the nuclear envelope, which remains intact during the whole process of cell division [9]. Bfa1 and Bub2, as well as the rest of SPB-associated MEN components, reside on the cytoplasmic side of the SPBs. However, signals that activate the GAP complex are also generated within the nucleus. As such, Bfa1/ Bub2 activity is required to maintain the functionality of the DNA damage checkpoint (DDC) and the spindle assembly checkpoint (SAC), 2 surveillance mechanisms that are respectively triggered by DNA lesions and the incorrect attachment of chromosomes to the mitotic spindle [4,10,11]. Hence, nucleocytoplasmic transport plays an important role in the regulation of the MEN. Transport across the nuclear envelope is mediated by nuclear pore complexes (NPCs), which are convoluted structures inserted in the nuclear membrane and organized into different subcomplexes of proteins named nucleoporins [12]. One of the modules that constitute the NPC in *S. cerevisiae* is the Nup82 complex, located at the cytoplasmic side and formed by the association of the Nup159, Nsp1, and Nup82 nucleoporins, which collaborate with Nup116, Nup42, Gle1, and Nup100 to facilitate nuclear mRNA export [12–14]. Notably, Nup159 has been identified as one of the proteins recognized by Atg8 to promote the autophagic degradation of NPC components [15,16]. Furthermore, the dynein light chain Dyn2 was recently identified as a novel constituent of the Nup82 complex that is recruited by Nup159 to the nuclear pores, which suggests that NPCs might also be important for the correct alignment of the mitotic spindle [17,18]. Therefore, similar to the MTOCs, NPCs are crucial elements in the regulation of many cellular processes besides their main function in nucleocytoplasmic transport.

Intriguingly, several connections have been established between the spindle MTOCs and the NPCs, which even share common components, suggesting that these structures collaborate in the regulation of key cellular processes [19]. Our results reveal a new link between proteins located on the SPBs and the NPCs. Specifically, we show that the Bfa1/Bub2 complex associates with the Nup159 nucleoporin. This interaction is cell cycle-regulated and requires Bfa1/Bub2 localization to the SPBs. Furthermore, we demonstrate that Bfa1/Bub2 association with Nup159 is prevented during the initial stages of spindle positioning but it is then promoted in anaphase to facilitate the activity of the Nup159-dependent autophagic pathway. The asymmetric localization of Bfa1/Bub2, which exclusively loads on the SPB that is delivered to the daughter cell [3], raises the interesting possibility that this novel connection between MEN components and Nup159 could mediate a differential regulation of the autophagic degradation of nucleoporins or other cellular components in the mother and daughter cells that might be important for the maintenance of the replicative lifespan in *S. cerevisiae*.

## Results

### A global screening reveals a novel interaction between nuclear pore components and the mitotic exit inhibitor Bfa1

In order to uncover yet undescribed proteins that could interact with Bfa1/Bub2 and regulate their function, we carried out a global screening using a two-hybrid assay and Bfa1 as the bait

[20]. Both Bub2 and the Cdc5 kinase, which phosphorylates and inactivates the Bfa1/Bub2 complex during anaphase [6], were identified among the proteins that associated with Bfa1 in our screening, demonstrating the validity of the approach. Interestingly, Nup159 and Nup42 were also found to interact with Bfa1 in the two-hybrid assay. These 2 FG-nucleoporins, characterized by phenylalanine- and glycine-rich sequences, localize to the cytoplasmic side of the nuclear pore and contribute to the formation of the filaments that project from this structure [12–14]. The cytoplasmic localization of Nup159 and Nup42 is in agreement with their potential interaction with the SPB-associated Bfa1/Bub2 complex. Moreover, since many of the signals that are transmitted to the GAP in order to prevent mitotic exit are generated within the nucleus, the interaction of these nucleoporins with Bfa1/Bub2 might represent a potential step mediating the communication between the nuclear compartment and the MEN inhibitors at the SPBs.

In order to verify the interaction between Bfa1 and the nucleoporins, we used co-immuno-precipitation assays. Indeed, 3HA-tagged Bfa1 was clearly pulled down together with green fluorescent protein (GFP)-labeled Nup159 in exponentially growing cells expressing both protein fusions, despite a residual background signal could sometimes be observed in control cells only expressing 3HA-Bfa1 due to unspecific binding of this protein to the magnetic beads used in the assay (Fig 1A). We also noticed that, independently of the epitope used for tagging, Nup159 is prone to degradation in protein extracts, which gives rise to several faster migrating bands in PAGE gels besides that of the full-length protein (Fig 1A). In contrast to what observed for Nup159, we could not co-immunoprecipitate Bfa1 together with Nup42 (S1A Fig), thus being unable to confirm their association with this assay. We also evaluated whether the confirmed Nup159 interaction with the GAP complex could depend on Nup42 expression. Deletion of the *NUP42* gene, however, did not impair the capacity of Nup159-GFP to pull down 3HA-Bfa1 in our assays (S1B Fig). Hence, we decided to not pursue the study of the possible Bfa1-Nup42 association any further.

The localization of Bfa1 and Bub2 to the SPBs is interdependent and the lack of Bub2 prevents Bfa1 phosphorylation, which likely takes place at this location [5,21]. Therefore, we next analyzed whether the Bfa1-Nup159 interaction was dependent on the integrity of the Bfa1/ Bub2 complex. Remarkably, despite showing similar levels of total protein in the initial extract, 3HA-Bfa1 did not efficiently co-immunoprecipitate with Nup159-GFP in *bub2*Δ cells (Fig 1A). In addition, Bub2-3HA could also be pulled down together with Nup159-GFP in co-immunoprecipitation assays, which indicates that this nucleoporin can associate with the whole GAP complex but not necessarily directly interact with both its components (Fig 1B). These results thus demonstrate that both Bfa1 and Bub2 associate with Nup159 and that an intact Bfa1/Bub2 complex is necessary for this interaction.

To provide further support to our observations, we also evaluated whether Nup159 and Bfa1 interacted in a bimolecular fluorescence complementation (BiFC) assay [22], which not only allows to detect the in vivo association between 2 proteins but also to determine where their interaction takes place. The BiFC is based in the reconstitution of the Venus yellow fluorescent protein by means of the association of 2 proteins that have been respectively fused to the N-terminal (VN) and C-terminal (VC) halves of this molecule [22]. Corroborating our prior results, Bfa1-VC interacted with Nup159-VN in the BiFC assay (Figs 1C and S1C). Remarkably, despite NPCs spreading all over the nuclear envelope, the association between Bfa1-VC and Nup159-VN was mainly restricted to the context of the SPBs, as demonstrated by colocalization of the BiFC signal with that of an mCherry-tagged version of the SPB component Spc42 (Fig 1C). The BiFC signal was faint and not detected in every cell (Fig 1C and 1D). Interestingly, Bfa1-VC also showed positive BiFC interaction with Dyn2-VN, another component from the Nup82 complex [13], and a limited association to Nup100-VN, which

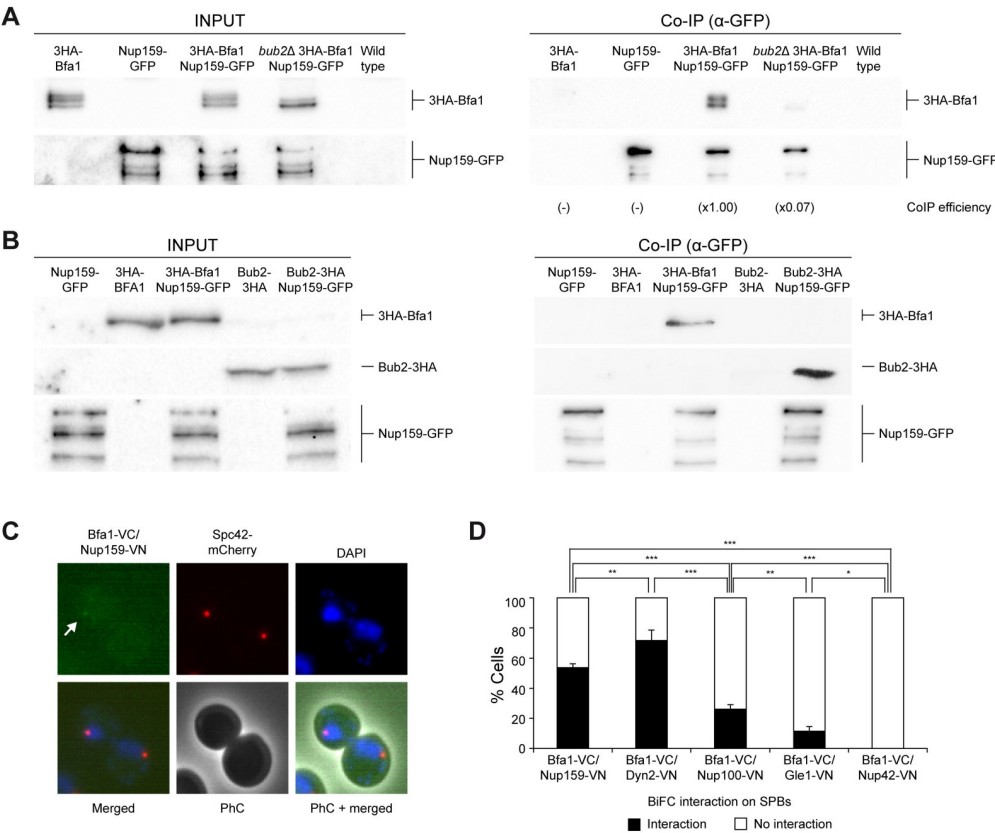

**Fig 1. The nucleoporin Nup159 is a novel Bfa1 interactor.** (A–D) Stationary phase cultures in YPAD were diluted to $OD_{600} = 0.2$ in fresh medium and then grown for 6 h at 26°C. (A, B) Co-immunoprecipitation analysis in cells simultaneously expressing Nup159-GFP and either 3HA-Bfa1 or Bub2-3HA in different genetic backgrounds. Cells exclusively expressing 3HA-Bfa1, Bub2-3HA, or Nup159-GFP were included as controls. Western blot gel images for 3HA-Bfa1, Bub2-3HA, and Nup159-GFP are shown for both the input (INPUT) and the immunoprecipitated (Co-IP) samples. The Co-IP efficiency for 3HA-Bfa1 relative to the corresponding control with untagged Nup159 (-) and referred to the strain used as a reference (×1.00) in (A) is indicated in each case. Each sample in (A) was separated from the rest with an empty well to discard any residual transfer between lanes. Experiments were carried out thrice ($n = 3$) and a representative image is shown. (C, D) BiFC analysis of Bfa1-VC interaction with VN-tagged nucleoporins. (C) Illustrative image displaying a positive BiFC interaction (Bfa1-VC/Nup159-VN, in green and marked with an arrow) and SPB localization (Spc42-mCherry, in red). Nuclear morphology (DAPI, in blue), PhC, and merged images are also shown. (D) Quantification of the percentage of cells displaying positive BiFC interaction. Data are the average of 3 samples ($n = 3$; 100 cells/each) and are available in S1 Data. Error bars represent SD. BiFC, bimolecular fluorescence complementation; GFP, green fluorescent protein; PhC, phase contrast; SPB, spindle pole body; VC, Venus fluorescent protein C-terminal halve; VN, Venus fluorescent protein N-terminal halve.

collaborates with Nup159-Nup82 [13] (Fig 1D). However, Bfa1-VC did not interact with Gle1-VN, a nucleoporin that more externally localizes in the cytoplasmic side of the NPC [12], or Nup42-VN (Fig 1D), in agreement with our previous observations (S1A Fig). Overall, these results demonstrate the interaction between Nup159 and Bfa1/Bub2 and support that it likely takes place in the context of the SPBs.

## The association of Bfa1 and Nup159 is cell cycle regulated

The Bfa1/Bub2 complex is posttranslationally modified and subjected to changes in its localization both as cells progress through the cell cycle and after activation of the mitotic checkpoints [5,6,21,23,24]. Therefore, we next analyzed whether the association of Nup159 with

Bfa1/Bub2 was modulated in a cell cycle-dependent manner. To this end, we synchronized cells in G1 with the α-factor pheromone and in metaphase or anaphase by means of the conditional inactivation of the thermosensitive *cdc13-1* or *cdc15-2* alleles, respectively [25–28]. Notably, the amount of 3HA-Bfa1 protein that was pulled down with Nup159-GFP in co-immunoprecipitation assays was particularly reduced in metaphase-arrested *cdc13-1* cells, especially when compared to anaphase-blocked *cdc15-2* cells (Fig 2A). The efficiency of the cell cycle arrest, which was confirmed in each case (S2A Fig), could be also easily verified by assessing the electrophoretic mobility of Bfa1, a protein that is unphosphorylated in G1 and gets progressively phosphorylated as cells go through mitosis, reaching its maximal phosphorylation level during anaphase [5].

In contrast to pheromone addition or inactivation of *cdc15-2*, expression of *cdc13-1* at the restrictive temperature restrains cell cycle progression due to the activation of a cell cycle checkpoint. Cdc13 is required to protect telomeres from degradation and, in its absence, cells accumulate single-stranded DNA at the chromosome ends, a signal that triggers a DDC-dependent metaphase arrest [25,26]. The reduced Nup159-Bfa1 association in *cdc13-1* cells at the restrictive temperature could thus be reliant either on cell cycle stage or on checkpoint activation. To discern between these 2 possibilities, we generated cells that expressed either the thermosensitive *cdc20-3* allele or, alternatively, an auxin-inducible degron of the anaphase-promoting complex (APC/C) cofactor Cdc20 (Cdc20-AID-9Myc) [29,30]. APC/C$^{Cdc20}$ elicits the metaphase-to-anaphase transition by promoting the proteasome-dependent degradation of both securin and the mitotic cyclins [31]. Hence, inactivation of *cdc20-3* or degradation of Cdc20-AID-9Myc impairs APC/C$^{Cdc20}$ activity and, consequently, blocks mitotic progression in metaphase without triggering any checkpoint [29,31]. Despite 3HA-Bfa1 co-immunoprecipitated more efficiently with Nup159-GFP in anaphase-arrested *cdc15-2* cells than after DDC activation in *cdc13-1* cells, their interaction was similarly reduced both in *cdc20-3* cells at the restrictive temperature and in Cdc20-AID-9Myc cells after auxin addition (Figs 2B and S2B). Hence, the decreased Nup159-Bfa1 association is due to the cell cycle stage and not to checkpoint activation. Accordingly, the BiFC interaction of Bfa1-VC and Nup159-VN also showed a cell cycle dependence, being less frequently observed in metaphase-arrested *cdc13-1* or *cdc20-3* mutants than in *cdc15-2* mutants blocked in anaphase (Fig 2C and 2D). Our results thus demonstrate that the interaction of Bfa1 and Nup159 is cell cycle-modulated, being their association specifically prevented during metaphase and strongly stimulated during anaphase.

## Neither the SAC nor the SPOC modulate the interaction between Nup159 and the Bfa1/Bub2 complex

Bfa1/Bub2 act as a central node that integrates signals from various checkpoints in order to inhibit mitotic exit [5,6,23]. In some instances, the signal that triggers the checkpoint is generated within the nucleus. This is the case for both the DDC, as discussed for *cdc13-1* cells, and the SAC, a surveillance mechanism triggered by unattached kinetochores [32]. Since Bfa1/Bub2 activity is essential for SAC functionality [6], we evaluated whether the association of Nup159 and Bfa1 was affected after cells were treated with the microtubule-depolymerizing agent nocodazole, which generates unattached kinetochores that activate the SAC and thus block cells in metaphase by preventing the Cdc20-dependent activation of the APC/C [32]. As Nup159 and Bfa1 interaction is reduced in metaphase, we used a *cdc20-3* background to fairly assess the effect of SAC activation on their association. Nocodazole treatment efficiently depolymerized spindle microtubules in *cdc20-3* cells that were previously arrested in metaphase at the restrictive temperature. However, no changes in the capacity of 3HA-Bfa1 to co-immunoprecipitate with Nup159-GFP were observed in nocodazole-treated or untreated cells (Fig

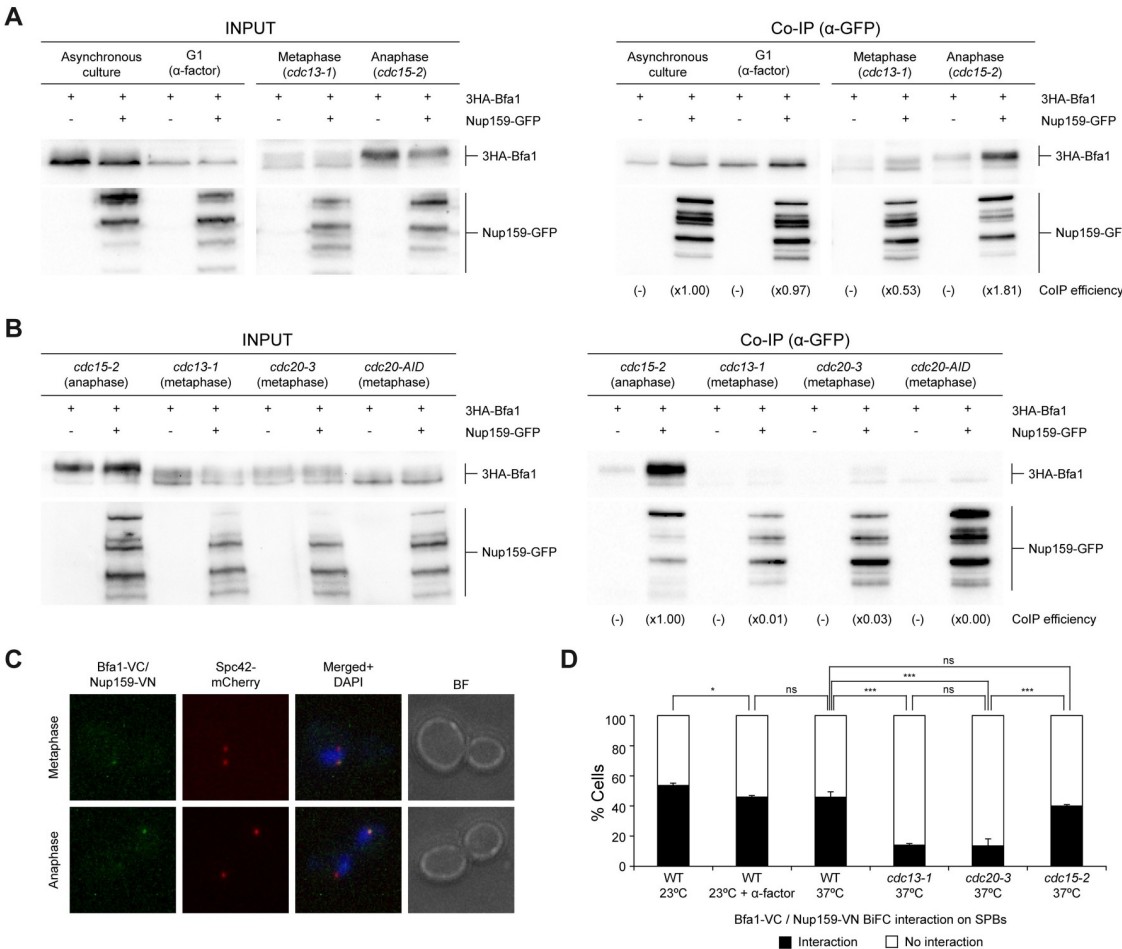

**Fig 2. Nup159-Bfa1 interaction is cell cycle regulated.** (A, B) Co-immunoprecipitation analysis in cells simultaneously expressing Nup159-GFP and 3HA-Bfa1 in the indicated genetic backgrounds. In each case, cells that only expressed 3HA-Bfa1 were included as controls. Western blot gel images for 3HA-Bfa1 and Nup159-GFP are shown for both the input (INPUT) and the immunoprecipitated (Co-IP) samples. The Co-IP efficiency for 3HA-Bfa1 relative to the corresponding control with untagged Nup159 (-) and referred to the strain or condition used as a reference (×1.00) is indicated in each case. (A) Stationary phase cells in YPAD were diluted to $OD_{600}$ = 0.2 in fresh medium and then grown for 6 h at 26˚C in YPAD (asynchronous culture) or arrested in G1 with 5 μg/ml α-factor in YPAD at 26˚C (G1 arrest) and, in the case of *cdc13-1* (Metaphase) and *cdc15-2* (Anaphase) cells, subsequently released for 2 h in YPAD at 34˚C. Experiment was carried out thrice (*n* = 3) and a representative image is shown. (B) Stationary phase cells in YPAD were diluted to $OD_{600}$ = 0.2 in fresh medium, arrested in G1 with 5 μg/ml α-factor and released for 2 h either in YPAD at 34˚C (*cdc13-1*, *cdc20-3*, and *cdc15-2* cells) or in YPAD with 500 μm IAA at 26˚C (*cdc20-AID* cells). (C, D) BiFC analysis of Bfa1 and Nup159 interaction. (C) Illustrative image displaying a positive BiFC interaction (Bfa1-VC/Nup159-VN, in green and indicated with an arrow) and SPB localization (Spc42-mCherry, in red). Nuclear morphology (DAPI, in blue), PhC, and merged images are also shown. (D) Quantification of the percentage of cells displaying positive BiFC interaction. Data are the average of 3 samples (*n* = 3; 100 cells/each) and are available in S1 Data. Error bars represent SD. AID, auxine inducible degron; BiFC, bimolecular fluorescence complementation; BF, bright-field; GFP, green fluorescent protein; IAA, auxine; SPB, spindle pole body; VC, Venus fluorescent protein C-terminal halve; VN, Venus fluorescent protein N-terminal halve; WT, wild type.

3A). Moreover, their association was not affected in cells further carrying a deletion of the *MAD2* gene, which encodes an essential SAC component [32] (Figs 3A and S2C). Hence, the Bfa1-Nup159 interaction is not modulated by the SAC in response to unattached kinetochores.

The SAC can be also activated due to problems in kinetochore integrity. A limiting step in the assembly of these structures is the loading of Ndc10, a structural component of the inner kinetochore region [33]. The thermosensitive *ndc10-1* allele encodes a mutant protein that

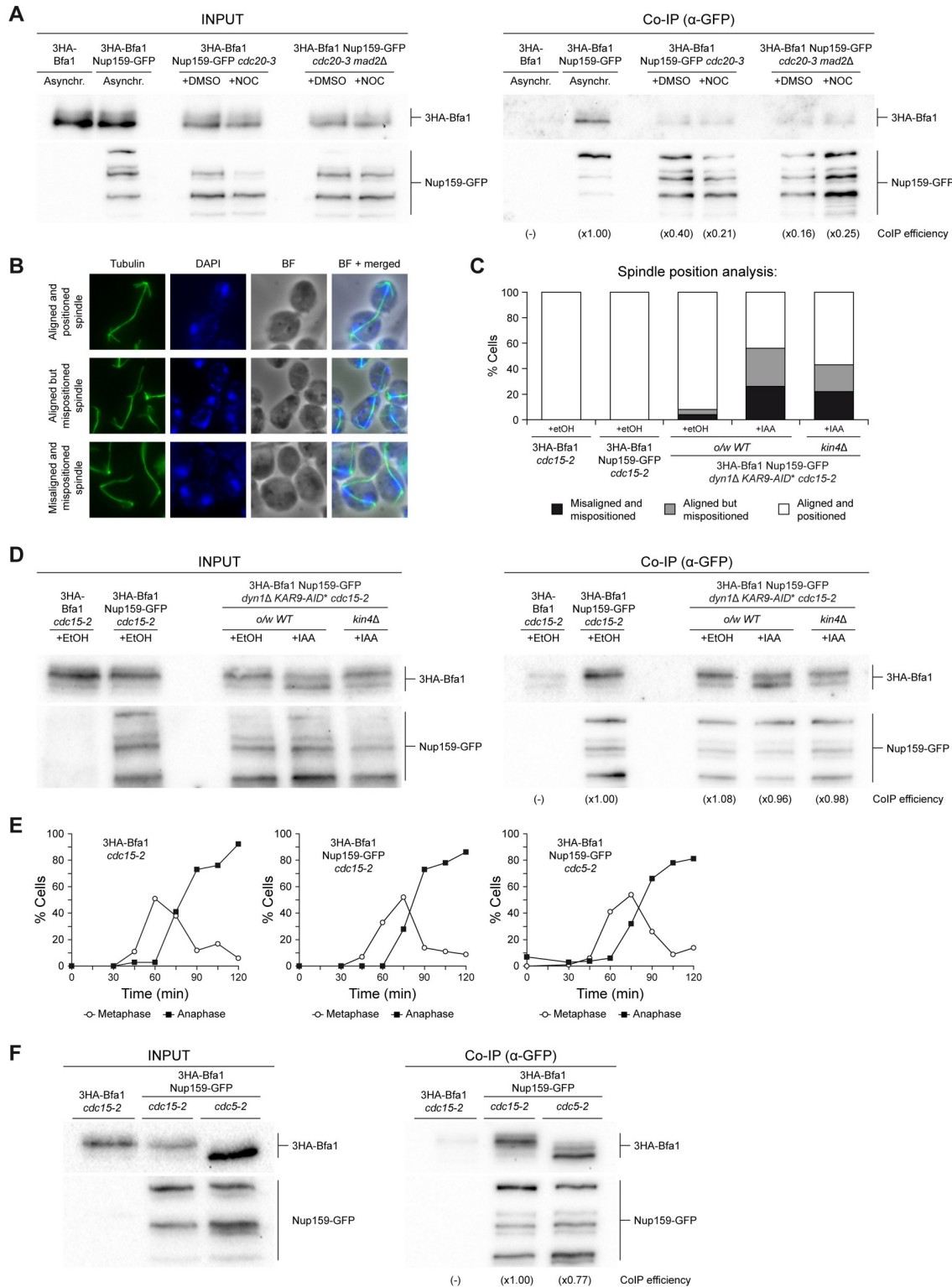

**Fig 3. Nup159 and Bfa1 interaction does not depend on the main mitotic checkpoints.** (A–F) Co-immunoprecipitation analysis in cells simultaneously expressing Nup159-GFP and 3HA-Bfa1 in the indicated genetic backgrounds. In each case, cells that only expressed 3HA-Bfa1 were included as controls. The Co-IP efficiency for 3HA-Bfa1 relative to the corresponding control with untagged Nup159 (-) and referred to the strain or condition used as a reference (×1.00) is indicated in each case (A, D, F). (A) Stationary phase cells in YPAD were diluted to OD$_{600}$ = 0.2 in fresh medium, arrested in G1 with 5 µg/ml α-factor and released for 2 h

into YPAD medium at 34°C without pheromone and with (+NOC) or without (+DMSO) 15 μg/ml nocodazole. Western blot gel images for 3HA-Bfa1 and Nup159-GFP are shown for both the input (INPUT) and the immunoprecipitated (Co-IP) samples. Experiment was carried out twice ($n = 2$) and a representative image is shown. (B–D) Stationary phase cells in YPAD were diluted to $OD_{600} = 0.2$ in fresh medium, arrested in G1 with 5 μg/ml α-factor and released for 2 h into YPAD medium at 34°C without pheromone and with (+IAA) or without (+EtOH) 2 mM auxin. (B, C) Spindle position analysis. (B) Illustrative immunofluorescence images of anaphase cells with mispositioned spindles, both correctly and incorrectly aligned, as well as of cells with properly aligned and positioned anaphase spindles. Tubulin (green), the nucleus (DAPI, blue), a BF and a merged image are shown. (C) Quantification of the percentage of cells in each of the previously established categories for anaphase spindle position. Data are available in S1 Data. (D) Western blot gel images for 3HA-Bfa1 and Nup159-GFP are shown for both the input (INPUT) and the immunoprecipitated (Co-IP) samples. Experiment was carried out thrice ($n = 3$) and a representative image is shown. (E, F) Stationary phase cells in YPAD were diluted to $OD_{600} = 0.2$ in fresh medium, arrested in G1 with 5 μg/ml α-factor and released into YPAD medium without pheromone at 34°C for 2 h. (E) Cell cycle progression according to spindle and nuclear morphologies. Percentages of metaphase and anaphase cells are indicated. Data are available in S1 Data. (F) Western blot gel images for 3HA-Bfa1 and Nup159-GFP are shown for both the input (INPUT) and the immunoprecipitated (Co-IP) samples. BF, bright-field; GFP, green fluorescent protein.

generates nonfunctional kinetochores and causes chromosome segregation errors [34]. We analyzed whether Nup159-Bfa1 association could be affected by activation of the SAC due to *ndc10-1* expression at the restrictive temperature. To facilitate comparison, cells further carried the *cdc20-3* allele. No differences were observed in the amount of 3HA-Bfa1 that co-immuno-precipitated with Nup159-GFP in metaphase-arrested *ndc10-1 cdc20-3* cells at the restrictive temperature when compared with otherwise wild-type *cdc20-3* cells, treated or not with noco-dazole (S2D and S2E Fig). These results further support that the association of Nup159 and the Bfa1/Bub2 complex is independent of the activation status of the SAC.

One last important surveillance mechanism that controls Bfa1/Bub2 activity is the SPOC [35]. The main SPOC effector is the Kin4 kinase, which phosphorylates Bfa1/Bub2 when the anaphase spindle is incorrectly positioned to prevent inactivation of the GAP complex by Cdc5 [7,8]. Additionally, Kin4 increases Bfa1/Bub2 dynamics on the SPB, which causes exclusion of the MEN-initiating GTPase Tem1 from this structure [24]. As a result, SPOC activation promotes the inhibition of MEN signaling [24,36]. Since Nup159-Bfa1 interaction is stimulated during anaphase, we evaluated a putative role of the SPOC in regulating their association. We used a genetic background in which the 2 pathways that position the mitotic spindle in budding yeast, the dynein- and the Kar9-dependent pathways [37–40], can be conditionally inactivated. Specifically, cells carried both a deletion of *DYN1*, the gene encoding the dynein heavy chain, and an auxin-inducible degron of Kar9 (Kar9-AID*-9Myc) [29,30]. Cells only displayed minor spindle position defects after *DYN1* deletion, since the 2 pathways can partially compensate for each other. However, additional inactivation of the Kar9 pathway generated severe spindle position problems (Fig 3B and 3C). As these defects strongly activate the SPOC [37,41], all strains further carried the *cdc15-2* allele to restrain cell cycle progression in anaphase and facilitate comparison. Remarkably, the induction of spindle misposition and subsequent SPOC activation did not significantly alter the capacity of 3HA-Bfa1 to co-immu-noprecipitate with Nup159-GFP (Figs 3D and S2F). Moreover, the levels of 3HA-Bfa1 that were pulled down with Nup159-GFP were not affected when spindle misposition was induced and activation of the SPOC was prevented as a consequence of the lack of Kin4 (Fig 3D). Hence, the SPOC does not regulate the association of Nup159 with the Bfa1/Bub2 complex in response to spindle alignment defects.

Cdc5 phosphorylates and inhibits Bfa1/Bub2 during anaphase to promote mitotic exit [6]. This kinase is a central target of the main mitotic checkpoints, which prevent Bfa1/Bub2 phos-phorylation by Cdc5 to restrain MEN signaling [5,6]. Inactivation of the thermosensitive *cdc5-2* allele blocks cells in anaphase but, in contrast to *cdc15-2*, maintaining the Bfa1/Bub2 complex in an unphosphorylated and active state [42] (Fig 3E and 3F). Remarkably, Nup159-GFP and

3HA-Bfa1 co-immunoprecipitated with the same efficiency in *cdc5-2* and *cdc15-2* cells arrested in anaphase at the restrictive temperature (Fig 3F). This result rules out that Cdc5 activity were necessary to promote the interaction between the nucleoporin and the GAP complex.

## Nup159 association with Bfa1 is reduced in the absence of Dyn2 and interferes with correct spindle positioning

Nup159 is part of the Nup82 subcomplex of the nuclear pore [43]. Interestingly, the yeast dynein light chain Dyn2, which is recruited by Nup159 to the nuclear pores, is another constituent of this subcomplex [18]. Nup159 structure is characterized by several well-defined domains, including an N-terminal β-propeller region that it is essential for nucleocytoplasmic mRNA transport, a central array of FG-rich repeat sequences, a dynein interaction domain (DID) that concentrates 5 consecutive Dyn2-binding motifs, and an α-helical C-region that plays an important role in nuclear pore anchoring, mRNA transport, and Nup159 protein stability [18,43–45]. To analyze whether the C-terminal domain of Nup159 could mediate its association with Bfa1, we used the *nup159-1* allele, which encodes a highly unstable protein lacking the last 96 aa of Nup159 [46]. Surprisingly, not only the interaction of Nup159 with Bfa1 was maintained in *nup159-1* cells, but their association seemed to be favored. Indeed, similar amounts of 3HA-Bfa1 co-immunoprecipitated with GFP-tagged versions of either wild-type Nup159 or the truncated nucleoporin lacking the C-domain, despite the latter being pulled down at lower levels (Fig 4A). Hence, the C-terminal domain is not necessary to establish Nup159 interaction with Bfa1, although it could be important to regulate the association between the nucleoporin and the GAP complex.

Mutation of K897 lysine in Nup159 is synthetically lethal with *KAR9* deletion, which is a characteristic feature of cells in which both the Kar9 and the dynein-dependent spindle-positioning pathways are disrupted. This suggests that the ability of Nup159 to target Dyn2 to the nuclear pores is important during spindle orientation [17]. To explore whether the Nup159-Bfa1/Bub2 interaction could be relevant for this process, and since Bfa1 associated with Dyn2 in our BiFC assays (Fig 1D), we next checked the capacity of Nup159 and Bfa1 to interact in the absence of Dyn2. Notably, immunoprecipitation analyses showed that the lack of Dyn2 resulted in a diminished ability of Nup159 to interact with Bfa1 (Fig 4B).

To better understand the functional relevance of the interaction between Nup159 and the Bfa1/Bub2 complex, we analyzed the consequences of forcing a constitutive association of Nup159 and Bfa1 throughout the cell cycle by using a GFP-binding protein (GBP)-based approach [47]. This method compels the interaction between 2 proteins by tagging one of them with GFP and the other with GBP, which selectively recognizes and strongly binds the green fluorescent molecule [47]. This approach further allows to evaluate the localization of the GFP-GBP complex by fluorescence microscopy. In cells expressing Bfa1-eGFP and Nup159-GBP, the green-fluorescent signal localized surrounding the nucleus and was not anymore restricted to the SPBs, which indicates that the constitutive interaction of both proteins drives a relocation of Bfa1-eGFP towards the nuclear pores (Fig 4C). The dynamics of Bfa1 exchange on the SPBs are important for the regulation of the cell cycle and the mitotic checkpoints [24,48]. Hence, we evaluated the consequences of a forced Nup159-Bfa1 interaction in a strain that, besides *BFA1-eGFP*, also carried an additional untagged copy of the *BFA1* gene. Analysis of cell cycle progression demonstrated that, after their release from a G1 arrest, cells expressing Bfa1-eGFP and Nup159-GBP showed a short but consistent 15 min delay at the metaphase-to-anaphase transition, which likely reflects problems at this cell cycle stage (Fig 4D–4G). This delay, similar to that caused by lack of the FEAR network-dependent Cdc14 release [49–51], was maintained up to anaphase, with spindle disassembly and mitotic exit

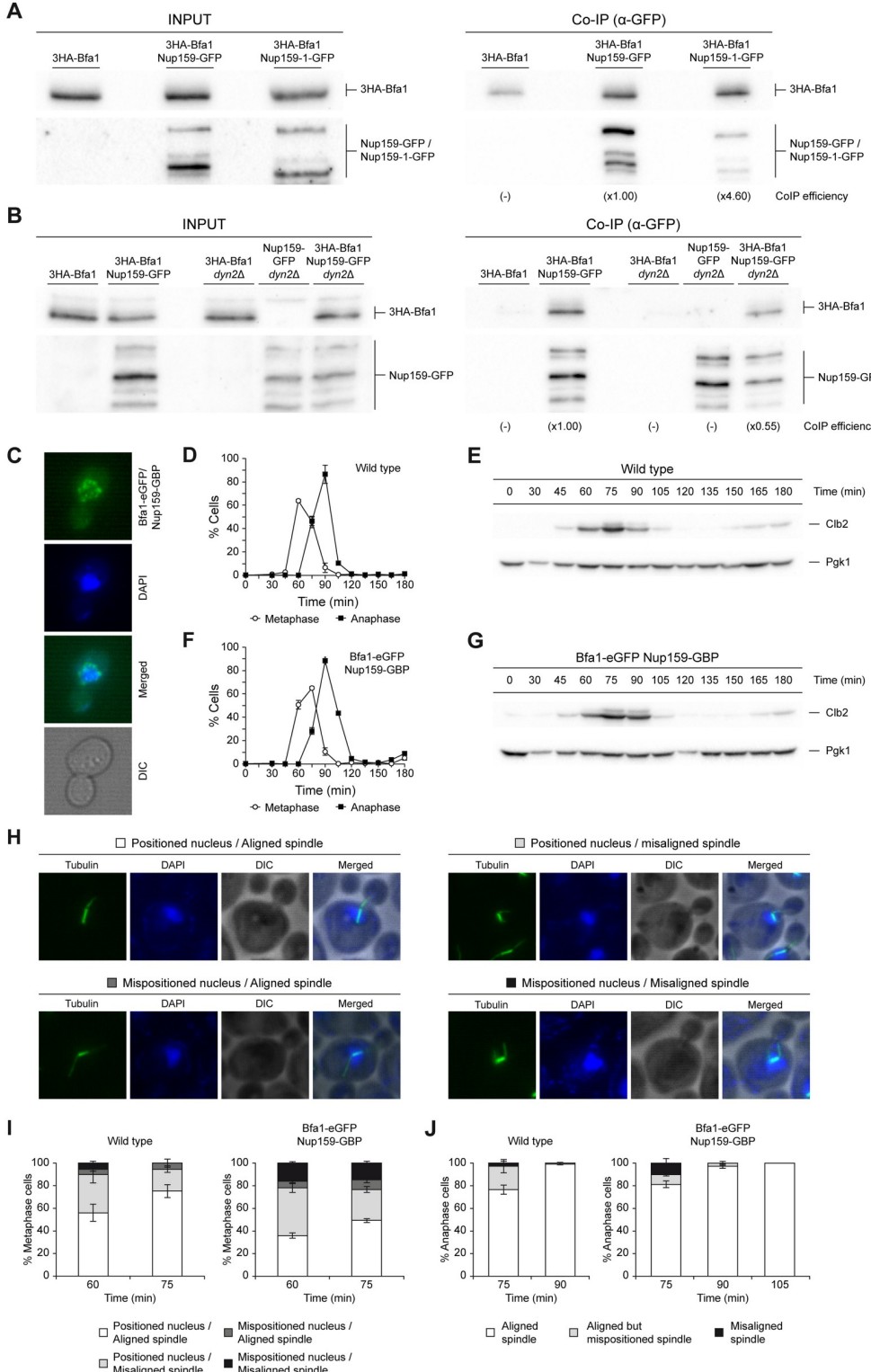

**Fig 4. Nup159 association with Bfa1 is reduced in cells lacking Dyn2 and interferes with early spindle positioning.**
(A, B) Co-immunoprecipitation analysis in cells simultaneously expressing Nup159-GFP and 3HA-Bfa1 in the
indicated strain backgrounds. In each case, cells that only expressed 3HA-Bfa1 were included as controls. Stationary
phase cultures in YPAD were diluted to $OD_{600} = 0.2$ in fresh medium and grown for 6 h at 23°C. Western blot gel
images for 3HA-Bfa1 and Nup159-GFP are shown for both the input (INPUT) and the immunoprecipitated (Co-IP)

samples. The Co-IP efficiency for 3HA-Bfa1 relative to the corresponding control with untagged Nup159 (-) and referred to the strain used as a reference (×1.00) is indicated in each case. (C–J) Stationary phase cultures of cells expressing an additional copy of *BFA1* integrated at the *URA3* locus, either alone or together with Bfa1-eGFP and Nup159-GBP protein fusions, were diluted to $OD_{600}$ = 0.2 in fresh YPAD medium, arrested in G1 with 5 μg/ml α-factor and released into YPAD medium without pheromone at 26°C. (C) Illustrative images of cells displaying Bfa1-eGFP (in green) and nuclear morphology (DAPI, in blue). DIC and merged images are also shown. (D–G) Cell cycle progression analysis. (D, F) Percentages of metaphase and anaphase cells according to spindle and nuclear morphologies. Data are the average of 3 samples (*n* = 3; 100 cells/each) and are available in S1 Data. Error bars represent SD. (E, G) Levels of Clb2 cyclin as determined by western blot analysis. Pgk1 was used as a control. (H–J) Spindle position analysis. (H) Illustrative immunofluorescence images of metaphase cells with a positioned or mispositioned nucleus, both displaying a correctly or incorrectly aligned spindle. Tubulin (green), the nucleus (DAPI, blue), a DIC, and a merged image are shown. (I) Quantification of the percentage of cells in each of the previously established categories for metaphase spindle position. (J) Percentage of anaphase cells with an aligned, aligned but mispositioned or misaligned spindle. Data are the average of 3 samples (*n* = 3; 50 cells/each) and are available in S1 Data. Error bars represent SD. DIC, differential interference contrast; GBP, GFP binding protein; GFP, green fluorescent protein.

taking place more slowly and gradually in cells expressing Bfa1-eGFP Nup159-GBP than in the wild type (Fig 4D–4G).

MEN components play an important role already during metaphase in regulating spindle positioning by controlling Kar9 localization [52]. Based on evidences linking Nup159, Dyn2, and Bfa1 with spindle alignment, we analyzed the consequences of forcing a constitutive Nup159-Bfa1 interaction on spindle and nuclear orientation. While most wild-type cells managed to correctly position the nucleus tangentially to the bud neck and aligned the metaphase spindle parallel to the mother-daughter cell axis already 75 min after release from an initial G1 (Fig 4H and 4I), simultaneous expression of Bfa1-eGFP and Nup159-GBP led to obvious defects in spindle orientation, as demonstrated by the accumulation of cells with a mispositioned nucleus and/or misaligned spindle at the same time point (Fig 4H and 4I). Nonetheless, the cells did finally manage to successfully position the mitotic spindle during late anaphase (Fig 4J). Our results thus demonstrate that the decrease in the association of Nup159 and Bfa1 in metaphase is important to facilitate a proper initial spindle alignment and nuclear positioning.

## A role for Bfa1 in Nup159-mediated autophagy

NPCs are specifically degraded both in a proteosome- and in an autophagy-dependent manner after cells are subjected to nitrogen starvation [16]. Interestingly, Nup159 was identified as one of the cargo-receptors that the core autophagy factor Atg8 recognizes and binds to facilitate loading of nucleoporins and/or nucleoporin complexes onto autophagosomes [15,16]. The *nup159-1* mutant displays aberrant phenotypes that include the presence of nucleoporin aggregates in the nuclear envelope even at the permissive temperature and a total lack of Nup159 protein at the restrictive temperature [53]. Although it might be alternatively explained by one of the binding partners being in saturation, the increased association of Bfa1 with the truncated Nup159-1 protein (Fig 4A) suggests that their interaction could be potentiated when nucleoporin aggregates accumulate and need to be cleared by autophagy. Furthermore, Nup159 also participate in other autophagic processes unrelated with the clearance of damaged nuclear pore components [54]. Hence, we finally explored the possibility that the Nup159-Bfa1 interaction could be relevant for autophagy.

After nitrogen starvation, selective autophagy of nucleoporins can be tracked by tagging these proteins with eGFP, since the compact fold of the green fluorescent molecule renders it resistant to vacuolar proteases, leading to an accumulation of eGFP in the cells [16]. Indeed, a reduction in the amount of full-length Nup159-eGFP and a subsequent increase in the total

intracellular levels of eGFP molecule could be observed after otherwise wild-type cells were transferred to medium lacking nitrogen (Fig 5A–5C and S3 Fig). Notably, a slight but consistent delay in the initiation of the autophagic degradation of full-length Nup159-eGFP after nitrogen deprivation was observed in cells lacking Bfa1 when compared to the wild type (Fig 5A and 5B). Accordingly, total levels of full-length Nup159-eGFP were initially higher in *bfa1Δ* cells than in the wild-type strain before nitrogen deprivation, suggesting an overall increased stability of this nucleoporin (Fig 5A and 5B). Furthermore, intermediate Nup159-eGFP degradation products accumulated more slowly in the *bfa1Δ* mutant after cells were transferred to medium lacking nitrogen (Fig 5A). These incomplete degradation forms of the nucleoporin originate in an autophagy-dependent manner, since they did not accumulate in wild-type or *bfa1Δ* cells carrying concurrent deletions of the *PEP4* and *PRB1* genes, which encode 2 key vacuolar proteases [16] (S4A Fig). Similarly, although the Nup133 nucleoprotein from the inner NPC core region was more resistant than Nup159 to autophagy induced by nitrogen starvation, degradation of Nup133-eGFP was also somewhat less efficient in a *bfa1Δ* mutant under these conditions (S4B Fig). This subtle defect in the autophagic degradation of Nup133-eGFP could be also evidenced by the accumulation of an intermediate degradation product when *bfa1Δ* cells were transferred to nitrogen-deprived medium (S4B Fig). Importantly, however, overall autophagy was normally induced in *bfa1Δ* cells after nitrogen depletion despite the initial defect in Nup159 degradation, as demonstrated by quantification of total levels of GFP-Atg8 and the accumulation of intracellular levels of GFP as a result of its autophagic degradation (Fig 5D). The previous data support that the autophagic clearance of Nup159-containing nucleoporin subcomplexes, although not severely compromised, seems to be somewhat obstructed in the *bfa1Δ* mutant, especially in early stages after nitrogen starvation. Accordingly, the lack of *BFA1* did not further enhance the defects in the autophagic degradation of Nup159 in cells expressing the *nup159-AIM* allele, which encodes a mutant nucleoporin that shows a reduced interaction with Atg8 [16] (S4C and S4D Fig). Likewise, a forced association between Nup159-AIM and Bfa1 using the GFP-GBP strategy was not able to rescue the defects of the AIM mutation (S4C and S4D Fig). Hence, the epistatic role of Bfa1 in the Atg8-dependent pathway cannot correct the defect in the association of Nup159-AIM with Atg8.

The modest defect in nucleoporin degradation observed in nitrogen-deprived *bfa1Δ* cells is not completely unexpected, since (i) nitrogen starvation causes an overall induction of autophagy processes and (ii) NPC clearance is not exclusively carried out in a Nup159-dependent manner. Recycling of NPCs, nucleoporins, and other nuclear components can also occur by other mechanisms, such as Atg39-mediated nucleophagy, piecemeal microautophagy, or ubiquitin-proteasome system [55]. Accordingly, cells treated with rapamycin became highly dependent on *BFA1* for their viability when they further accumulated damaged NPCs due to expression of the *nup159-1* allele, even at the permissive temperature (Fig 5E). Moreover, the defect in cell viability after rapamycin treatment was similar in *nup159-1 bfa1Δ* and *nup159-1 atg39Δ* cells (Fig 5F). Hence, we next analyzed autophagy when damage to nuclear pores was selectively generated using the *nup159-1* allele. Specific activation of autophagy in *nup159-1* cells was verified by an increase in *ATG8* gene expression (S4E Fig). In agreement with our hypothesis, and in contrast to what observed in nitrogen-deprived cells (Fig 5D), GFP-Atg8 degradation was less efficient in a *bfa1Δ* mutant when autophagy was induced by NPC damage caused by *nup159-1* expression, as observed by a reduced accumulation of free GFP (Fig 6A–6C). Accordingly, GFP-Atg8 foci that formed as a consequence of *nup159-1* expression accumulated more efficiently in the vacuoles of *atg15Δ* cells, where this autophagy factor is directed for degradation, than in an *atg15Δ bfa1Δ* mutant (Fig 6D and 6E). Deletion of *ATG15*, which encodes a lipase that is indispensable for dissolving autophagosomal membranes in the

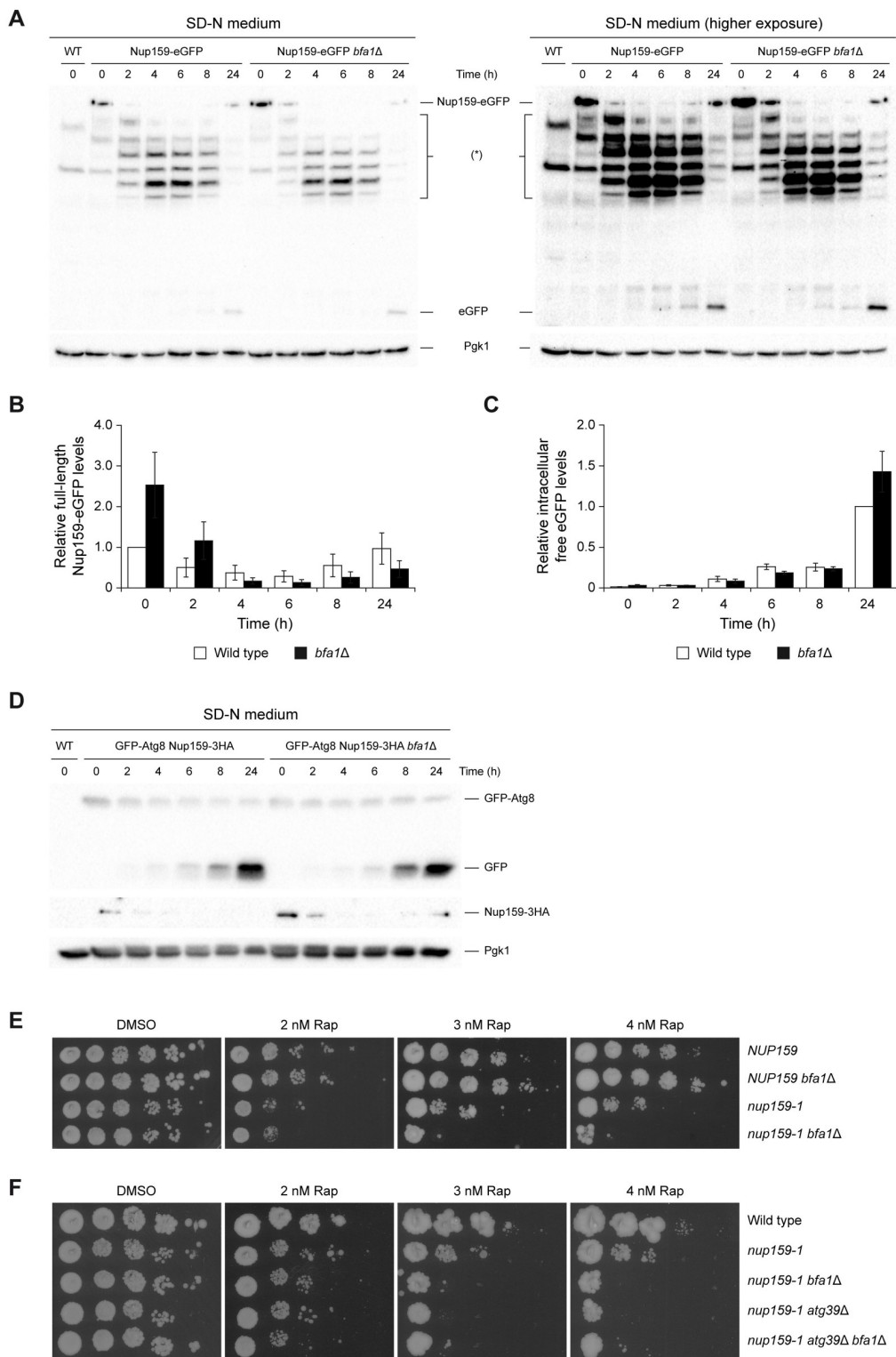

**Fig 5. A role for Bfa1 and Nup159 in autophagy.** (A–D) Stationary phase cultures in YPAD were diluted to $OD_{600}$ = 0.2 in SD-N medium and grown for 24 h at 26°C. (A) Western blot gel images displaying Nup159-eGFP and free eGFP levels at the indicated time points after cells were transferred to SD-N medium (time = 0 h). Intermediate Nup159-eGFP degradation forms are indicated with (*). Pgk1 was used as a loading control. To facilitate visualization of the fainter bands overexposed images of the same gels are also shown (higher exposure). Experiment was carried out

thrice (*n* = 3) and a representative image is shown. (B, C) Quantification of the levels of full-length Nup159-eGFP (B) and free eGFP (C) in western blot experiments. Data are the average of 3 experiments (*n* = 3) and are available in S1 Data. Error bars represent SEM. (D) Western blot gel images displaying GFP-Atg8, free GFP, and Nup159-3HA levels at the indicated time points after cells were transferred to SD-N medium (time = 0 h). Pgk1 was used as a loading control. Experiment was carried out twice (*n* = 2) and a representative image is shown. (E, F) Cells were plated by spotting tenfold serial dilutions of an exponential liquid culture ($OD_{600}$ = 0.5) on YPAD medium with the indicated concentrations of rapamycin (Rap) and cultured at 26˚C. A control without rapamycin (DMSO) was also included. Experiment was carried out thrice (*n* = 3) and a representative experiment is shown. GFP, green fluorescent protein.

vacuole, was introduced to facilitate the visualization of autophagic degradation intermediates [16]. Notably, no additive defect in GFP-Atg8 degradation was found in *nup159-1 bfa1Δ* cells after the Atg39-dependent pathway was impaired (S4F and S4G Fig). This, together with the fact that the simultaneous deletion of *ATG39* and *BFA1* did not cause a synergistic defect in viability (Fig 5F), suggests that Atg39-dependent nucleophagy is not the main pathway that substitutes for the lack of Bfa1.

To reinforce our results, we also followed the delivery of membrane-embedded nucleoporin complexes to autophagosomes after cells expressing Nup192-eGFP were starved of nitrogen. Autophagy activation leads to an accumulation of foci of the eGFP-tagged nucleoporin in the vacuole [16]. Visualization of NPC degradation intermediates was again facilitated either by using an *atg15Δ* mutant background or, alternatively, strains lacking the Rab family GTPase Ypt7, which is required for the fusion of autophagosomes with the vacuole [56] (S5A and S5B Fig). Notably, *BFA1* deletion led to a significant decrease in the number of vacuolar Nup192-eGFP foci, supporting a role of Bfa1 in autophagy (S5A and S5B Fig).

NPCs and nucleoporin subcomplexes targeted by autophagy form clusters at specific sites of the nuclear envelope, sometimes causing invagination or protrusion of the nuclear membrane. Nup159-containing clusters are cleared from the nuclear envelope in an Atg8-dependent manner and are subsequently sent to the vacuole [16]. Autophagic clearance of these clusters after nitrogen starvation can be followed by the accumulation of vacuolar Nup159-eGFP fluorescent foci [16,57]. Remarkably, targeting of membrane-embedded Nup159-eGFP foci was impaired in the absence of Bfa1, as evidenced by a strong reduction in the number (and intensity) of extranuclear Nup159-eGFP foci as well as the subsequent increase in the quantity of perinuclear foci that cannot be cleared from the nuclear envelope and, consequently, a higher percentage of cells displaying nuclear membrane aberrations (Fig 6F–6I). Importantly, these foci increased in a *ypt7Δ* mutant and were not observed in an *atg8Δ* background (S5C–S5F Fig), which demonstrates that they are bona fide markers for autophagic bodies. These results strongly support that Bfa1 participates in the Nup159-dependent autophagic pathway.

## Discussion

The nuclear pores are formed by large protein complexes that allow the exchange of molecules between the nucleus and the cytoplasm. The main components of NPCs are nucleoporins, a group of proteins that show a high degree of conservation throughout evolution [12]. A cumulative body of evidences demonstrates that NPCs fulfill many other roles in the cells, from gene expression control and transcriptional processing to ensuring the activity and functionality of the mitotic checkpoints [12]. Interestingly, recruitment of an NPC to the preexistent SPB is also required for the duplication of this MTOC and the insertion of the newly generated SPB into the nuclear envelope [58]. We have established a novel link between NPCs and SPBs by unveiling the interaction between the Nup159 nucleoporin and the SPB-associated Bfa1/Bub2

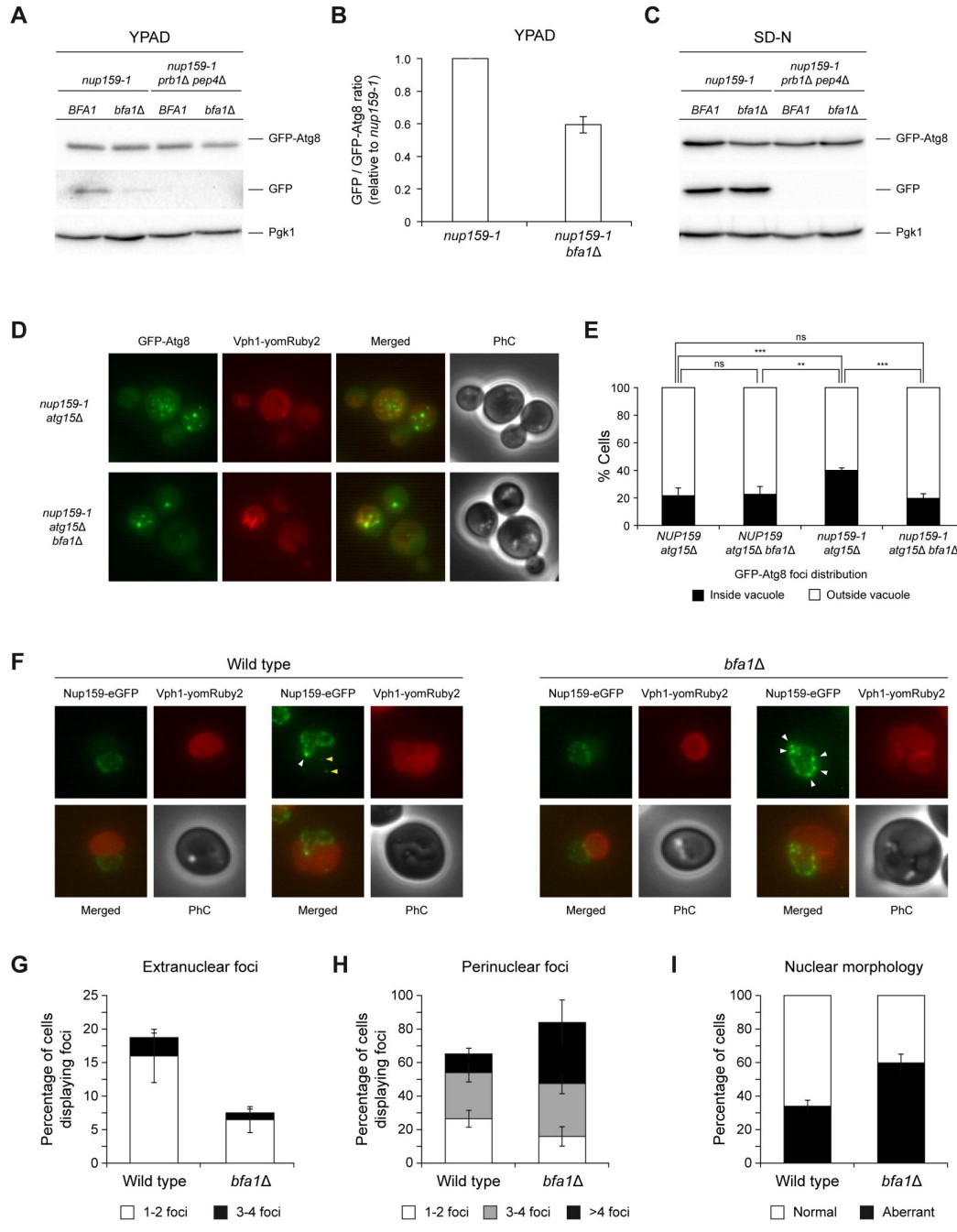

**Fig 6. Bfa1 specifically collaborates with Nup159 in the autophagic clearance of nucleoporin complexes.** (A–I) Exponential cultures in YPAD were diluted to $OD_{600} = 0.2$ in YPAD (A, B), SC (D, E), or SD-N medium (C, F–I), and grown for 4 (D–I) or 24 h (A–C) at 26˚C. (A, C) Western blot gel images displaying GFP-Atg8 and free GFP levels 24 h after cells were diluted in YPAD (A) or SD-N medium (C). Pgk1 was used as a loading control. Experiment was carried out thrice ($n = 3$) and a representative image is shown. (B) Quantification of the relative levels of free GFP in (A). Data are the average of 3 experiments ($n = 3$) and are available in S1 Data. Error bars represent SEM. (D) Representative images of live cells expressing GFP-Atg8 (green) and Vph1-yomRuby2 (red) in *nup159-1 atg15Δ* cells, lacking or not Bfa1. PhC and merged images are also shown. (E) Quantification of the percentage of cells displaying (black bars) or not (white bars) GFP-Atg8 foci inside the vacuole. Data are the average of 3 experiments ($n = 3$; 100 cells/each) and are available in S1 Data. Error bars represent SD. (F) Illustrative images of live cells expressing Nup159-eGFP (green) and Vph1-yomRuby2 (red) both in a *bfa1Δ* and in and otherwise wild-type background. Perinuclear and extranuclear Nup159-eGFP foci are indicated with white and yellow arrows, respectively. PhC and merged images are also shown. (G–I) Quantification of the percentage of cells displaying extranuclear (G) and perinuclear (H) Nup159-eGFP foci, as well as of cells displaying an aberrant nuclear

morphology (I). An estimation of the percentage of cells displaying 1–2, 3–4, or more than 4 foci is also shown for both cells with extranuclear (G) and perinuclear (H) Nup159-eGFP clusters. Data are the average of 4 experiments (*n* = 4; 100 cells/each) and are available in S1 Data. Error bars represent SEM. GFP, green fluorescent protein; PhC, phase-contrast.

complex, which inhibits mitotic exit signaling in *S. cerevisiae* and constitutes a central target of the main cell cycle checkpoints in this organism [21,59,60].

Nup159 belongs to the FG subgroup of nucleoporins and is a member of the Nup82 complex [43,45]. We have identified this nucleoporin in a global two-hybrid screening for yet-unknown proteins that interacted with Bfa1 [21,59,60]. The Nup159-Bfa1 association, further substantiated by co-immunoprecipitation and BiFC analyses, requires an intact Bfa1/Bub2 complex and it is likely to take place when the GAP is loaded on the SPBs. Although another nucleoporin, Nup42, was also originally identified in our screening as a potential Bfa1 interactor, we could not finally confirm their in vivo association. This, however, supports that the Nup159-Bfa1 interaction is specific and not the result of a promiscuous association of Bfa1/Bub2 with FG nucleoporins. Lack of Nup42, on the other hand, did not disrupt the interaction of Nup159 with the GAP complex either. It is worth noting that, unlike Nup159, Nup42 is not an essential protein. The fact that the structure and functionality of the Nup82 subcomplex is not greatly affected by the lack of Nup42 could thus explain why the Nup159-Bfa1 association is maintained in its absence.

The interaction of Nup159 with Bfa1/Bub2 is cell cycle regulated, being less favored during metaphase and strongly stimulated later in anaphase. Metaphase is also the cell cycle stage at which the turnover of the GAP complex on the SPBs is more dynamic [24]. Hence, the reduced Nup159-Bfa1 association at metaphase might in principle be explained based on a lower residence time of Bfa1/Bub2 on the SPBs and consequently to a diminished capacity to interact with Nup159 at this location, where their association likely takes place. However, dynamics of Bfa1/Bub2 loading on the SPBs are also perturbed after the SPOC is triggered, since Kin4 phosphorylation actively excludes the GAP from these structures [24], and a strong Bfa1-Nup159 interaction was still observed in anaphase-arrested cells under these conditions. This result firmly supports that the reduced association of Nup159 and Bfa1/Bub2 during metaphase and the subsequent increase in their interaction later in anaphase are cell cycle-regulated events that are subjected to a specific control. In anaphase, Bfa1/Bu2 phosphorylation by the Polo-kinase Cdc5 is a key event that inactivates the GAP complex, thereby allowing MEN signaling and mitotic exit [6]. However, the activity of this kinase is not necessary to promote the strong interaction between the GAP and Nup159 in anaphase. This result was nonetheless somehow expected, since Cdc5 activity is minimal in G1, and the Nup159-Bfa1 association is still evident at this cell cycle stage.

Bfa1/Bub2 also plays a pivotal role in the maintenance of genome integrity and a correct ploidy [5,6,23]. The DDC, the SAC, and the SPOC all depend on an active Bfa1/Bub2 complex to maintain their functionality, despite acting at different cell cycle stages and being triggered by distinct events [5,6,23]. Furthermore, DDC and SAC are activated by a signal in the nucleus that must be transmitted to the GAP complex, which resides at the cytoplasmic side of the SPBs. Hence, Nup159 represents a plausible candidate to channel checkpoint signaling from the nucleus towards Bfa1/Bub2. However, the Nup159-Bfa1 interaction is not regulated by the activation of these surveillance mechanisms and thus, although we cannot completely rule out this possibility, our results do not support a role of Nup159 in checkpoint signaling.

Spindle positioning in *S. cerevisiae* depends on the Kar9 and dynein pathways, which can partially compensate for each other [37–39]. The dynein motor complex is formed by heavy

(Dyn1), intermediate (Pac11 and Dyn3), and light (Dyn2) chains [40,61]. Interestingly, Dyn2 is also a component of the Nup82 complex [18]. The capacity of Nup159 to target Dyn2 to the nuclear pores was proposed to play a role in the dynein-mediated process of spindle orientation and nuclear segregation [17]. Supporting this idea, a *nup159K897R* mutant displays a synthetic spindle position defect with the deletion of *KAR9*, indicating that ubiquitylation of Nup159 in K897 is important for the functionality of the dynein-dynactin pathway [17]. Our results now show that a forced interaction of Nup159 and Bfa1 generates problems during the early stages of spindle alignment. Accordingly, a permanent association of Nup159 and Bfa1 causes a delay at the metaphase-to-anaphase transition. This observation is in agreement with the reduction in the Nup159-Bfa1 association during metaphase. The newly uncovered interaction with Bfa1/Bub2 could thus extend the relevance of Nup159 in the process of spindle and nuclear orientation. NUP133, a nucleoporin that belongs to the Nup107–160 complex (the largest NPC subcomplex in higher eukaryotes), facilitates efficient anchoring of the dynein/dynactin complex to the nuclear envelope, which contributes to centrosome positioning [62]. This link between NPC components and the centrosome in humans suggests an evolutionary conservation that highlights the relevance of the studies aiming to better understand the functional role of their connection. Notably, tethering of centrosomes to the nuclear envelope by an NPC-mediated dynein/dynactin-dependent anchoring at the G2/M transition contributes to the initial stages of bipolar spindle assembly [62]. This agrees with our results suggesting a combined role of Nup159 and the Bfa1/Bub2 complex during the initial steps of mitotic spindle alignment. Noteworthy, despite temporarily affecting initial spindle assembly and positioning, disruption of the link between the NPC and the centrosome in human HeLa cells is eventually overcome by additional mechanisms that allow the final establishment of a bipolar spindle [62]. The same is true when the interaction of Nup159 and the Bfa1/Bub2 complex is forced in budding yeast cells, since constitutive Nup159-Bfa1 association only induces a similar delay in the metaphase-to-anaphase transition than that described for FEAR mutants (approximately 15 min), a nonessential mitotic exit-promoting pathway in budding yeast [51]. However, despite the subtle defect in cell cycle progression under normal growth conditions, a coordinated role of these proteins could be required under certain adverse situations. Accordingly, the connection between the NPC and the centrosome in human cells was proposed to be more relevant in oocytes, non-rounding cells (HeLa cells experience strong cell rounding at mitotic entry), or under pathological conditions [62].

Interestingly, besides its essential role in nucleocytoplasmic transport, Nup159 acts as a cargo receptor for autophagy. Defective or unassembled Nup159-containing nucleoporin complexes are recognized by the core factor Atg8 to be directed to autophagosomes [15,16]. Nup159 thus serves as an element that controls for the integrity or function of the NPCs [15,16,63]. Additionally, Nup159 also participates in other autophagic-dependent processes, such as the Snx4-assisted vacuolar targeting of certain transcription factors [54]. Notably, autophagic degradation of Nup159 under nitrogen starvation is less efficient in the absence of Bfa1. Furthermore, while their role is likely taken over by alternative pathways when autophagy is globally induced in the cells, the collaborative function of Nup159 and Bfa1 in autophagy becomes highly important when aggregates of damaged NPCs are specifically generated. The interaction of Nup159 and Bfa1 is likely required to facilitate early steps of the autophagic process. Accordingly, a strong accumulation of Nup159 clusters that cannot be cleared from the nuclear envelope and later targeted to the vacuole is observed early after autophagy is induced in cells lacking Bfa1. These clusters presumably represent Nup159-containing nucleoporin complexes, since the Nup159-Bfa1 association is required not only for an efficient autophagic degradation of Nup159, but also of Nup133 and Nup192. This phenotype is similar to that observed in the absence of the Nup116 nucleoporin, which disrupts Atg8 binding to

Nup159 and, consequently, the autophagic clearance of NPCs, leading to the formation of herniae in the nuclear envelope [57]. Bfa1 might therefore act as an adaptor that facilitates the interaction of Nup159 with Atg8 when misassembled NPCs and/or damaged nucleoporin complexes accumulate, thus promoting their autophagic degradation. Similarly, Bfa1 might also assist Nup159 in other autophagic processes that are mediated by this nucleoporin.

As budding yeast ages, cells accumulate misassembled NPCs that do not contribute to overall transport kinetics. These damaged NPCs specifically lack a set of FG-Nups that decline during aging [64]. Furthermore, defective NPCs are restricted from being transmitted to daughter cells during asymmetric divisions to prevent aging [65]. Notably, in *S. cerevisiae*, NPCs are not randomly distributed, but organized in clusters that concentrate around the SPBs [66]. Hence, the association of Nup159 with Bfa1/Bub2 might contribute to prevent the inheritance of dysfunctional NPCs in the daughter cell in different ways. Firstly, the interaction of asymmetrically localized SPB components with FG-Nups that decay with age might help ensuring that functional NPCs are preferentially inherited by the daughter cell. Accordingly, Nup159 and Bfa1 association is stimulated after metaphase, when the Bfa1/Bub2 complex is already predominantly localized to the SPB that finally segregates into the bud. On the other hand, their interaction might further favor the preferential inheritance of functional NPCs in the daughter cell by modulating the autophagic degradation of misassembled nucleoporin complexes that were nonetheless eventually transported into the bud. In this way, Bfa1 association with Nup159 might promote the Atg8-dependent autophagic degradation of Nup159-containing dysfunctional NPCs, thereby constituting a backup quality control mechanism that limited the amount of old and damaged NPCs that are nonetheless still received by the newly duplicated cell.

The Nup159-Bfa1 interaction is not the first link described between components of the NPC and the MEN pathway. Indeed, deletion of *NUP1*, which encodes a nucleoporin of the NPC nucleoplasmic side, is synthetically lethal with an allele of the MEN gene *NUD1* that carries a missense mutation (*nud1-G585E*) [19]. Furthermore, both *nup1Δ bfa1Δ* and *nup1Δ bub2Δ* cells are inviable, but *nup1Δ* does not display genetic interactions with proteins acting downstream of Tem1, suggesting that this functional link is limited to components that act early in the MEN pathway [19]. Our results and these previous evidences demonstrate that interactions are limited to specific nucleoporins and MEN proteins, and not the result of a generic association between NPC and SPB components [19]. Nup159 is the yeast homolog of human nucleoporin NUP214. Repression of NUP214 by ectopic expression of miR-133b, a miRNA down-regulated in head and neck squamous cell carcinoma, delays mitotic progression in HCT116 cells [67]. Interestingly, NUP214 also associates to the spindles during mitosis, although the role that it could be playing at this location is still unknown [68,69]. Furthermore, despite no direct links have been so far established between NUP214 and autophagy, a fusion involving NUP214 and the sequestosome-1 (SQSTM1) protein, which is required for proper autophagy induction, has been connected with acute lymphoblastic leukemia [70]. Hence, our results could contribute to a better understanding of the functional connections between NPCs and the spindle MTOCs and how defects in their concerted activities can be at the origin of human diseases.

## Materials and methods

### Strains and plasmids

All strains are W303 derivatives and are listed in S1 Table, which also indicates the strains used in each figure. Strains carrying GFP-, eGFP-, mCherry-, and yomRuby2-tagged fusion proteins were generated by amplifying the corresponding tag sequences using previously

described primers [71,72]. Subsequently, the amplification products were integrated by homologous recombination at the C-terminus of the gene, before the stop codon. Strains for BiFC analyses were constructed following an analogous approach [22]. Finally, a similar strategy was used for gene deletion, but the endogenous locus was replaced with a cassette carrying a selectable marker [73].

## Cell culture

Cells were grown in YPAD (YP (1% yeast extract, 2% peptone) with 2% glucose and 300 µg/ml adenine), SC (0.17% yeast nitrogen base, 0.5% ammonium sulfate, 2% glucose, 0.2% Drop-out mix), or SD-N (0.17% yeast nitrogen base without amino acids and ammonium sulfate, 2% glucose) medium. Experiments normally started with stationary phase cultures in YPAD medium that were diluted to optical density at 600 nm ($OD_{600}$) = 0.2 in fresh medium. For the analysis of cells in asynchronous cultures, cells were subsequently grown in YPAD for 6 h at 26°C or 34°C. For synchronous cell cycle analyses, the diluted stationary cultures were arrested in G1 with 5 µg/ml α-factor and then released into fresh YPAD medium without pheromone and grown at 26°C or 34°C. For autophagy experiments, diluted stationary cultures were instead grown for 2 h in YPAD medium and then transferred to SD-N medium and grown for up to 24 h at 26°C.

## Fluorescence microscopy

Fluorescently tagged proteins and DAPI (4′, 6-diamidino-2-phenylindole) staining for nuclear analysis were visualized as described in [74]. A DM6000 microscope (Leica) equipped with a 100×/1.40 NA (numerical aperture) oil immersion objective and a DFC350 FX digital charge-coupled device camera (Leica) was used to image the cells. The obtained images were processed and analyzed with LAS AF (Leica) and ImageJ (http://rsbweb.nih.gov/ij/) software.

## Immunofluorescence

Immunofluorescence for the analysis of cell cycle progression was performed as meticulously detailed in [49], using specific antibodies at the concentrations described in S2 Table. Samples were analyzed and imaged as indicated for visualizing the fluorescently tagged proteins.

## Protein extraction and western blot analysis

Protein extracts were prepared using a trichloroacetic acid (TCA) precipitation method detailed in [50]. For TCA precipitation, 10 ml cells from liquid culture were incubated for 10 min in 5% TCA. Samples were centrifuged for 3 min at 1,400 rcf and 4°C, and pellets were washed, transferred to clean tubes, and resuspended in 1 ml acetone at room temperature using a vortex mixer. Samples were next centrifuged for 7 min at 1,400 rcf, and the collected pellets were dried in a hood and resuspended in 125 µl lysis buffer [50 mM Tris-HCl (pH 7.5), 1 mM EDTA, 50 mM DTT, 1 mM PMSF, complete EDTA-free protease inhibitor cocktail (Roche)]. After addition of an equal volume of glass beads, cells were lysed in a vortex mixer for 40 min at 4°C. Finally, 62.5 µl 3× Laemmli sample buffer was added, and protein extracts were boiled for 5 min at 100°C before being loaded in a polyacrylamide gel. Western blot analysis of protein levels was performed as described in [50], using specific antibodies at the concentrations indicated in S2 Table. The protein expression levels were detected and quantified using WesternBright ECL reagents (Advansta), a ChemiDoc MP system, and Image Lab software (Bio-Rad).

## Protein co-immunoprecipitation

For co-immunoprecipitation assays, 50 ml exponential yeast culture ($OD_{600}$ = 0.8) were harvested and washed once in 1 ml cold water. Cells were then centrifuged at 11,000 ×$g$ and 4˚C and either immediately processed or alternatively frozen in liquid $N_2$. Pellets were next resuspended in 500 μl lysis buffer [50 mM Tris-HCl (pH 7.5), 250 mM NaCl, 10% glycerol, 10 mM EDTA (pH 8.0), 1 mM DTT, 0.5 mM PMSF, 1× complete EDTA-free protease inhibitor cocktail (Roche)] and then lysed using a Multi-beads shocker (Yasui Kikai Corporation) for 40 min at 4˚C, alternating 60-s pulses at 2,500 rpm with 60-s rest. The extracts were cleared twice by centrifugation at 500 ×$g$ for 5 min at 4˚C to eliminate cell debris. Then, Triton X-100 was added to make up 0.5% final concentration, and the extracts were incubated at 4˚C with rotation for 90 min. After detergent treatment, the extracts were centrifuged twice at 11,000 ×$g$ and 4˚C for 15 min, and the supernatant was transferred to new tubes. Protein concentration was adjusted by measuring the absorbance at 280 nm with a NanoDrop system (Thermo Scientific) or with a Bradford assay. An equal amount of protein extracts (at least 3,000 μg of protein) was adjusted to a total volume of 1 ml in solubilization buffer [50 mM Tris-HCl (pH 7.5), 250 mM NaCl, 10% glycerol, 10 mM EDTA (pH 8.0), 0.5% Triton X-100] and processed for immunoprecipitation. Additionally, 100 μg of each protein extract were also saved for the input samples and stored at −20˚C. For immunoprecipitation, 50 μl GFP-Trap magnetic micro-beads (μMACS, Miltenyi Biotec) were added to the samples and incubated for 30 min at 4˚C. Subsequently, the samples were transferred to columns that had been previously equilibrated in 200 μl solubilization buffer, using magnets to retain the GFP-Trap beads. Columns were washed 4 times with solubilization buffer and once with 100 μl μMACS washing buffer [20 mM Tris-HCl (pH 7.5)]. In order to separate the protein from the beads, columns were incubated with 20 μl of previously boiled μMACS elution buffer (Miltenyi Biotec) for 5 min at room temperature, after which 50 μl of the same buffer were additionally added. The immunoprecipitated protein samples were transferred to clean tubes. In parallel, input samples were adjusted to a total volume of 50 μl with solubilization buffer, after which the same volume of 3× Laemmli buffer with 6% β-mercaptoethanol was added to each tube. Both input and immunoprecipitated were warmed for 5 min before undergoing SDS-PAGE. Western blot analysis of the protein levels was performed as detailed in [50], using specific antibodies at the concentrations indicated in S2 Table. The protein expression levels were detected and quantified using WesternBright ECL reagents (Advansta) and a ChemiDoc MP system (Bio-Rad) or the ImageJ (http://rsbweb.nih.gov/ij/) software.

## Quantification of gene expression

For quantitative RT-PCR analyses, 10 ml of culture were centrifuged and resuspended in 400 μl of TES buffer (10 mM Tris-HCl (pH 7.5), 10 mM EDTA, 0.5% SDS). An equal volume of phenol was added and samples were first incubated at 65˚C for 45 min, then at 4˚C for 5 min, and finally centrifuged for 5 min at 13,000 g and 4˚C. The resulting aqueous phase was similarly processed again, first with an equal volume of phenol, and one last time with an equal volume of chloroform. The final aqueous phase was mixed with 40 μl of 3 M Sodium Acetate (pH 5.2) and 1 ml of ethanol, and precipitated for 1 h at −20˚C. After centrifugation of the samples, the pellet containing the RNA was washed with 70% ethanol, dried, and finally resuspended in 50 μl diethylpyrocarbonate-treated $H_2O$. RT-PCR reactions were performed in a 7500 Real-Time PCR System (Applied Biosystem) using 2 μg of total RNA. The RNA was first treated with DNase I (Invitrogen) and then retrotranscribed to cDNA using the SuperScript III Reverse Transcriptase kit (Invitrogen). Quantitative PCRs were carried out using a 1:5 dilution of the cDNA sample, iTaq Universal SYBR Green Supermix and the primers shown in S3

Table. Ct values and the Ct mean for the different replicates were obtained using the 7500 Real-Time PCR Software v2.06.

## Statistics and reproducibility

Statistical details for each experiment, including the specific measure used to estimate the variation within each group of data (SD or SEM), the number of times that the experiments have been independently repeated and the exact value of *n* in each case, are given in the figure legends. In all experiments, control samples were always treated as the problem.

## Supporting information

**S1 Fig. Analysis of Nup159 and Nup42 interaction with Bfa1.** (A, B) Co-immunoprecipitation analysis in cells simultaneously expressing 3HA-Bfa1 and Nup159-eGFP, both in a *nup42Δ* or in an otherwise wild-type background, as well as in cells from another strain that concurrently expresses 3HA-Bfa1 and Nup42-eGFP. Cells expressing only 3HA-Bfa1, Nup159-eGFP, or Nup42-eGFP, as well as the wild-type strain, were also included as controls. Stationary phase cultures in YPAD were diluted to $OD_{600} = 0.2$ in fresh medium and grown for 6 h at 26°C. Western blot gel images for 3HA-Bfa1, Nup159-eGFP, and/or Nup42-eGFP are shown for both the input (INPUT) and the immunoprecipitated (Co-IP) samples. The Co-IP efficiency for 3HA-Bfa1 relative to the corresponding control with untagged Nup159 (-) and referred to the strain used as a reference (×1.00) in (B) is indicated in each case. (C) Gray scale images for each of the individual fluorescent channels in Fig 1C, which displays a positive BiFC interaction between Bfa1-VC and Nup159-VN.
(TIF)

**S2 Fig. Nup159-Bfa1 interaction depends on cell cycle stage but not checkpoint activation.** (A, B) Percentage of cells in metaphase, anaphase or other stages of the cell cycle, for the co-immunoprecipitation experiments shown in Fig 2A (A) and 2B (B). Data are available in S1 Data. (C) Percentage of cells that did not display microtubules, as well as those of cells in metaphase, anaphase, or other stages of the cell cycle, for the co-immunoprecipitation experiment shown in Fig 3A. Data are available in S1 Data. (D, E) Co-immunoprecipitation analysis in cells simultaneously expressing 3HA-Bfa1 and Nup159-GFP in a *cdc20-3*, a *cdc20-3 ndc10-1* or an otherwise wild-type background. Cells expressing 3HA-Bfa1 were included as a control. Stationary phase cells in YPAD were diluted to $OD_{600} = 0.2$ in fresh medium and either grown in YPAD medium at 26°C for 6 h (Asynchr.) or alternatively arrested in G1 with 5 μg/ml α-factor and then released into YPAD medium at 34°C without pheromone and with (+NOC) or without (+DMSO) 15 μg/ml nocodazole. (D) Percentage of cells that did not display microtubules, as well as those of cells in metaphase, anaphase, or other stages of the cell cycle. Data are available in S1 Data. (E) Western blot gel images for 3HA-Bfa1 and Nup159-GFP for both the input (INPUT) and the immunoprecipitated (Co-IP) samples. The Co-IP efficiency for 3HA-Bfa1 relative to the corresponding control with untagged Nup159 (-) and referred to the strain or condition used as a reference (×1.00) is indicated in each case. (F) Percentage of cells in metaphase, anaphase, or other stages of the cell cycle, for the co-immunoprecipitation experiment shown in Fig 3B–3D. Data are available in S1 Data.
(TIF)

**S3 Fig. Autophagic degradation of Nup159 in cells deprived of nitrogen.** (A) Western blot gel images of the three biological replicates used for the quantifications in Fig 5B and 5C, displaying Nup159-eGFP and free eGFP levels at the indicated time points after cells were transferred to SD-N medium (time = 0 h). Intermediate Nup159-eGFP degradation forms are

indicated with (*). Pgk1 was used as a loading control. Graphs showing the quantification of the levels of full-length Nup159-eGFP and free eGFP for each of the experiments are also included next to each western blot image. Data are available in S1 Data.
(TIF)

**S4 Fig. Role of Bfa1 and Nup159 in autophagy after cells are deprived of nitrogen.** (A–D) Stationary phase cultures in YPAD were diluted to $OD_{600}$ = 0.2 in SD-N medium and grown for 24 h at 26˚C. (A) Western blot gel images displaying Nup159-eGFP and free eGFP levels at the indicated time points are shown for otherwise wild-type, *NUP159-eGFP* and *NUP159-eGFP bfa1Δ* cells, all further carrying *PEP4* and *PRB1* gene deletions, after being transferred to SD-N medium (time = 0 h). Intermediate Nup159-eGFP degradation forms are indicated with (*). Pgk1 was used as a loading control. To facilitate visualization of the fainter bands overexposed images of the same gels are also shown (higher exposure). Experiment was carried out thrice (*n* = 3) and a representative image is shown. (B) Western blot gel images displaying Nup133-eGFP and free eGFP levels at the indicated time points are shown for wild type, *NUP133-eGFP* and *NUP133-eGFP bfa1Δ* cells after being transferred to SD-N medium (time = 0 h). Intermediate Nup133-eGFP degradation forms are indicated with (*). Pgk1 was used as a loading control. To facilitate visualization of the fainter bands overexposed images of the same gels are also shown (higher exposure). Experiment was carried out thrice (*n* = 3) and a representative image is shown. (C) Western blot gel images displaying Nup159-eGFP in wild-type cells, as well as levels of Nup159-AIM-eGFP in cells expressing Bfa1-GBP, in a *bfa1Δ* mutant or in an otherwise wild-type background, 24 h after being transferred to SD-N medium. Pgk1 was used as a loading control. Experiment was carried out thrice (*n* = 3) and a representative image is shown. (D) Quantification of the relative levels of free eGFP in (C). Data are the average of 5 experiments (*n* = 5) and are available in S1 Data. Error bars represent SEM. (E) *ATG8* gene expression determined by quantitative RT-PCR in the indicated strains and normalized to the wild type. Data are the average of 3 experiments (*n* = 3) and are available in S1 Data. Error bars represent SEM. (F) Western blot gel images displaying GFP-Atg8 and free GFP levels 24 h after exponential cells were diluted in SD-N medium. Pgk1 was used as a loading control. Experiment was carried out 4 times (*n* = 4) and a representative image is shown. (G) Quantification of the relative levels of free GFP in (F). Data are the average of 4 experiments (*n* = 4) and are available in S1 Data. Error bars represent SEM.
(TIF)

**S5 Fig. Autophagic clearance of nucleoporin complexes is disrupted in cells lacking Bfa1.** (A–F) Stationary phase cultures in YPAD were diluted to $OD_{600}$ = 0.2 in SD-N medium and grown for 4 h (C–F) or 24 h (A, B) at 26˚C. (A) Representative images of live cells expressing Nup192-eGFP (green) and Vph1-yomRuby2 (red) in *atg15Δ* and *atg15Δ bfa1Δ* cells. Phase-contrast (PhC) and merged images are also shown. (B) Quantification of the percentage of cells displaying (black bars) or not (white bars) extranuclear Nup192-eGFP foci. Data are the average of 3 experiments (*n* = 3; 100 cells/each) and are available in S1 Data. Error bars represent SD. (C–E) Quantification of the percentage of cells displaying extranuclear (C) and perinuclear (D) Nup159-eGFP foci, as well as of cells displaying an aberrant nuclear morphology (E), 4 h after being transferred to SD-N medium. An estimation of the percentage of cells displaying 1–2, 3–4, or more than 4 foci is also shown for both cells with extranuclear (C) and perinuclear (D) Nup159-eGFP clusters. Data are the average of 3 experiments (*n* = 3; 100 cells/each) and are available in S1 Data. Error bars represent SEM. (F) Quantification of the percentage of cells displaying (black bars) or not (white bars) extranuclear Nup159-eGFP foci. Data are the average of 3 experiments (*n* = 3; 100 cells/each) and are available in S1 Data. Error

bars represent SD.
(TIF)

**S1 Table. Strains.** List of the strains in this study, which also details the specific experiment in which each of the strains was used.
(DOCX)

**S2 Table. Antibodies for immunofluorescence and western blot.** List of the antibodies used in this study, both for immunofluorescence and western blot analyses.
(DOCX)

**S3 Table. Oligonucleotide sequences for quantitative RT-PCR.** List of primers used in this study for the analysis of gene expression by quantitative RT-PCR.
(DOCX)

**S1 Raw Images. Original images for blots.** Original images of all blots displayed in this study.
(PDF)

**S1 Data. Numeric data.** Raw numeric data used in this study.
(XLSX)

# Acknowledgments

We thank members of the Monje-Casas' laboratory for critical reading of the manuscript and Dr. Hélène Gaillard for her useful suggestions. We also thank Drs. A. Amon, Charles N. Cole, and M. Muñiz for generous gifts of plasmids, strains, and/or additional material.

# Author Contributions

**Conceptualization:** Fernando Monje-Casas.

**Data curation:** Inés García de Oya, Javier Manzano-López, Alejandra Álvarez-Llamas, María de la Paz Vázquez-Aroca, Cristina Cepeda-García, Fernando Monje-Casas.

**Formal analysis:** Inés García de Oya, Javier Manzano-López, Alejandra Álvarez-Llamas, María de la Paz Vázquez-Aroca, Cristina Cepeda-García, Fernando Monje-Casas.

**Funding acquisition:** Fernando Monje-Casas.

**Investigation:** Inés García de Oya, Javier Manzano-López, Alejandra Álvarez-Llamas, María de la Paz Vázquez-Aroca, Cristina Cepeda-García.

**Methodology:** Inés García de Oya, Javier Manzano-López, Alejandra Álvarez-Llamas, María de la Paz Vázquez-Aroca, Cristina Cepeda-García, Fernando Monje-Casas.

**Supervision:** Fernando Monje-Casas.

**Validation:** Inés García de Oya, Javier Manzano-López, Alejandra Álvarez-Llamas, María de la Paz Vázquez-Aroca, Cristina Cepeda-García, Fernando Monje-Casas.

**Visualization:** Inés García de Oya, Javier Manzano-López, Alejandra Álvarez-Llamas, María de la Paz Vázquez-Aroca, Cristina Cepeda-García.

**Writing – original draft:** Fernando Monje-Casas.

**Writing – review & editing:** Fernando Monje-Casas.

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
