## [Editor Report · Decision Letter 0]

23 May 2022

Dear Dr Monje-Casas, 

Thank you for submitting your manuscript entitled "Association of the Nup159 nucleoporin with asymmetrically-localized spindle pole body proteins facilitates selective autophagy of nuclear pore components." for consideration as a Research Article by PLOS Biology.

Your manuscript has now been evaluated by the PLOS Biology editorial staff as well as by an academic editor with relevant expertise and I am writing to let you know that we would like to send your submission out for external peer review.

Once your full submission is complete, your paper will undergo a series of checks in preparation for peer review. After your manuscript has passed the checks it will be sent out for review. To provide the metadata for your submission, please Login to Editorial Manager (https://www.editorialmanager.com/pbiology) within two working days, i.e. by May 25 2022 11:59PM.

Kind regards,

Ines

--

Ines Alvarez-Garcia, PhD

Senior Editor

PLOS Biology

---

## [Decision Letter · Decision Letter 1]

12 Jul 2022

Dear Dr Monje-Casas,

Thank you for your patience while your manuscript entitled "Association of the Nup159 nucleoporin with asymmetrically-localized spindle pole body proteins facilitates selective autophagy of nuclear pore components." was peer-reviewed at PLOS Biology, and please accept my apologies for the delay. Your manuscript has been evaluated by the PLOS Biology editors, an Academic Editor with relevant expertise, and by two independent reviewers.

The reviews are attached below. As you will see, the reviewers find the conclusions potentially interesting, but they also raise a substantial number of concerns that should be addressed before we can consider the manuscript further for publication. Both reviewers suggest several experiments that should be performed to confirm your findings. After consulting wth the Academic Editor, it is clear that a substantial amount of work would be required to meet the criteria for publication in PLOS Biology. However, we would be open to inviting a comprehensive revision of the study that thoroughly addresses all the reviewers' comments.

Given the extent of revision that would be needed, we cannot make a decision about publication until we have seen the revised manuscript and your response to the reviewers' comments. Your revised manuscript would need to be seen by the reviewers again, but please note that we would not engage them unless their main concerns have been addressed.

We appreciate that these requests represent a great deal of extra work, and we are willing to relax our standard revision time to allow you 6 months to revise your study. Please email us (plosbiology@plos.org) if you have any questions or concerns, or envision needing a (short) extension.

**IMPORTANT - SUBMITTING YOUR REVISION**

3. Resubmission Checklist

a) *PLOS Data Policy*

b) *Published Peer Review*

Sincerely,

Ines

--

Ines Alvarez-Garcia, PhD

Senior Editor

PLOS Biology

Reviewers' comments

Rev. 1:

"Association of the Nup159 nucleoporin with asymmetrically-localized spindle pole body-associated proteins facilitates selective autophagy of nuclear pore components" by Fernando Monje-Casas and coworkers describes the discovery of a physical interaction between the nuclear pore complex (NPC), nucleoporin 159 in particular and the Bfa1/Bub2 complex of spindle-pole bodies in budding yeast. The paper finds that the interaction is cell cycle regulated and dependent on the yeast Polo-like kinase Cdc5. Functionally, the authors aim to link this interaction to the previously identified role of Nup159 as receptor for autophagic degradation of the NPC and show delays in mitotic progression upon forcing protein interaction.

Overall, the investigation of the molecular interaction between Nup159 and Bfa1 is done thoroughly and detailed and can be convincing pending further technical improvements. The functional part, linking Bfa1 to autophagic degradation of NPCs is, however, underdeveloped and will require substantial further experimental work to support the authors claims.

Major points:

Nup159- Bfa1 interaction:

1 - Figure 1A-B: Comparison of different samples are done across different gels, which is not acceptable. Also, the manuscript would be strengthened by quantitative and not qualitative comparisons.

2 - Throughout the manuscript, the authors use different cell cycle mutants to induce arrests. They need to (a) provide data that cell cycle arrest worked as expected and (2) since cell cycle arrest often involves temperature shift also clearly indicate that control cells have been treated the same.

3 - Figure 3A: Input for Nup159 has a poor quality and should be repeated.

4 - Figure 4B: I do not agree with the conclusion, since the authors also see less Nup159 in

the input of dyn2Δ compared to the other cases.

5- Figure 5: The interpretation of this "synthetic interaction experiment" is unclear and is not put very well in the context of the functional analysis in the next figures. Since currently, the authors do not have a loss-of-function/loss-of-interaction mutant, it may be advisable to leave these experiments out of the current paper and include in a future manuscript OR strengthen the link to the analysis of NPC autophagy.

Role in Autophagic degradation of NPCs:

6 - Figure 6A-C: Why are Nups completely degraded after 2 h SD-N treatment? This is in contrast to previous studies and should be discussed. Importantly, the phenotype in the bfa1-mutant is very mild and not apparent after 24h. Given this, individual replicates of experiments need to be shown both as qualitative western blots and in the quantification, where mean +- SD is not acceptable any longer. Also, conclusions need to be toned down given the mild effect. In the same vein, Supplement Figure 3B shows actually no difference in the degradation of nup133 +/- Bfa1. This is somehow unexpected since NPCphagy should degrade Nup133 as well.

7 - I suggest that the authors should also investigate Nup degradation under the same

conditions they see the strongest interaction between Bfa1 and Nup159. Moreover,

they should test whether the nup159-AIM mutant has a similar effect.

8 - Interpretation of Figure 6D is unclear. Who says that these foci are autophagic

bodies / autophagosomes. I suggest to do the same assay in atg15Δ. Moreover, one could delete Atg8,1,7 or 5 to see if the foci disappear.

Additional points:

9 - They started with a Y2H screen but none of the data are available. They should

provide at least the data for the case they report. In general, I would suggest to

publish the results of the whole yeast-two hybrid screen.

10 - For all western blots size markers are not displayed throughout the manuscript.

11 - Labelling of the IP data is not very clear. It should be indicated in the figure against

which tag the IP was performed.

12 - The statement "The GFP-GBP methodology has the additional advantage that allows evaluation of the localization of the protein complex by fluorescence microscopy." Is misleading since the the GFP-GBP methodology works by forcing two proteins together bringing with it artificial localization of the complex.

Rev. 2:

This paper by Garcia de Oya et al. centers on the discovery of a putative physical interaction between the spindle pole body localized Bfa1/Bub2 complex and the nucleoporin Nup159 in budding yeast. Much of the manuscript explores how this interaction might be impacted by cell cycle cues and there is the suggestion that it may function in spindle positioning and nuclear pore complex (NPC) autophagy upon nitrogen starvation. Overall, the data are heavily reliant on co-immunoprecipitations/Western blots that vary in quality (e.g. relative amounts/stability of proteins in western blots are inconsistent between experiments) and it remains unknown whether the interaction between Nup159 and Bfa1 is actually direct. Furthermore, there is no clear sense of what the ultimate function of the interaction is as there are no experiments that specifically disrupt it. There is, however, an attempt to force the interaction by using a GBP-GFP approach but these data are difficult to interpret. Thus, the paper is preliminary and does not provide a significant advance to our understanding of NPC phagy or spindle positioning beyond the novel link between Nup159 and Bfa1.

Thoughts for the authors should they wish to improve technical aspects of the paper:

1) Many of the Western blots presented in this manuscript are of inconsistent quality and lack quantification. For example, the amount of 3HA-Bfa1 that was co-IP'ed with Nup159-GFP in Figure 1A looks to be much more than the amount in Figure 2A, although these should be identical experiments. In addition, although the authors claim that the interaction between Bfa1 and Nup159 is reduced in metaphase, the amount of Bfa1 that co-IP'd with Nup159-GFP in Figure 3B looks to be similar to the amount in 2A from asynchronous and G1 arrested cells. There is also no way to assess the efficiency of the IP without knowing the fraction of the input and bound fractions loaded on the gels.

2) The idea that the Bfa1-Nup159 interaction promotes the degradation of NPCs by autophagy is not well supported by the data and is open to alternative explanations. It is also conceptually challenging how an interaction at the SPB could lead to degradation of NPCs. In terms of the presented data, the authors suggest that degradation of Nup159 (and Nup133) is delayed in the absence of Bfa1. This might be the case but the upstream signaling driving autophagy would have to be more thoroughly explored as there does not appear to be any major change in the kinetics of degradation per se. Indeed, based on their blot in Figure 6A, the degradation kinetics of Nup159 look identical with or without Bfa1. According to their quantification of the relative amount of free GFP in Figure 6C, it seems like there could be slightly less degradation at 6 hours of nitrogen starvation in bfa1Δ cells compared to wildtype, however, comparing this to the relative amount of full-length protein from Figure 6B (which is also reduced compared to wildtype), it is unclear if there would be any evidence of reduced degradation if the free GFP band was normalized to the full length Nup159-GFP band (as is the standard method of quantifying autophagic degradation).

3) The bimolecular fluorescence complementation experiment in Figure 1C could be supported by additional specificity controls (e.g. additional nups and SPB components) and perhaps the incorporation of the temperature sensitive mutants that are thought to prevent the interaction (metaphase arrested) or promote the interaction (anaphase arrested).

4) Please include blots to show the degradation of Cdc20-AID and Kar9 in the presence of IAA.

5) Do cells in Figure 3 B-D include the Kar9-AID-9Myc construct and dyn1Δ as indicated from the call out in the text? If so, please fix. If not, it is unclear what the IAA drug is doing.

6) Please include genotypes of the cells shown in Figure 3B.

---

## [Decision Letter · Decision Letter 2]

13 Mar 2023

Dear Dr Monje-Casas,

Thank you for your patience while we considered your revised manuscript entitled "Association of the Nup159 nucleoporin with asymmetrically-localized spindle pole body proteins facilitates selective autophagy of nuclear pore components." for publication as a Research Article at PLOS Biology. Your revised study has been evaluated by the PLOS Biology editors, the Academic Editor and the two original reviewers.

You will see that the reviewers appreciate the improvements you have made in the manuscript, however they remain sceptical of the role of the Nup159 and Bfa1 interaction in nuclear pore clearance by autophagy. They suggest further experiments that could be performed to confirm this or to consider alternative explanations. In light of the reviews (attached below), we would like to invite you to revise the work to address the remaining points of the reviewers.

**IMPORTANT - SUBMITTING YOUR REVISION**

3. Resubmission Checklist

a) *PLOS Data Policy*

b) *Published Peer Review*

Sincerely,

Ines

--

Ines Alvarez-Garcia, PhD

Senior Editor

PLOS Biology

Reviewers' comments

Rev. 1:

"Association of the Nup159 nucleoporin with asymmetrically-localized spindle pole body-associated proteins facilitates selective autophagy of nuclear pore components" by Fernando Monje-Casas and coworkers describes the discovery novel physical interactions between components of the nuclear pore complex (NPC) and spindle-pole bodies in budding yeast.

Overall, I find the revised version of the paper improved in some parts and I think the investigation on the molecular interactions is worth to be published after including necessary controls. However, I am not convinced that the data allows to link these interactions with NPC quality control. Therefore, this claim/interpretation should be removed from the paper.

Major points:

Nup159- Bfa1 interaction:

1 - Figure 1 - Please support your argument in the rebuttal letter by showing full images of western blots. Also, please include Molecular Weight markers.

2 - Figure S1a - How can you conclude specificity with two GFP-tagged proteins?

3 - Figure 1B - "immunoprecipitationassays, which indicates that this nucleoporin is able to interact with both components of the GAP complex (Figure 1B)." This statement is wrong, rephrase "Nup159 can interact with the GAP complex but not necessarily directly interact with both components."

4 - Figure 1C - include grey scale image for the single channels. PhC + merged can be removed since there is no additional information gained from it.

5 - Cell cycle regulation - Yes, cell cycle arrests in budding yeast are often more efficient compared to other systems. The authors argument that they worked to 100% in all strains in all experiments is however hard to believe. Please include the according quantifications in the manuscript. There is enough room in the supplementary data.

6 - "Interestingly, Bfa1-VC also showed positive BiFC interaction with Dyn2-VN, another nucleoporin from the Nup82 complex" Dyn2 is not a nucleoporin, rephrase to "component"

7 - Figure 4A - Co-IPs are difficult to evaluate if bait is not pulled down to equal amounts. The binding partner may already be in saturation.

9 - Figure 5B,C - Include statistics / biological replication to show significance.

Role in Autophagic degradation of NPCs:

10 - Figure 6A-C: The main conclusion of this part of the paper are not backed up by experiments. An alternative explanation may be an mRNA export defect of nup159-1. Induction of autophagy needs, however, increased expression of Atg8 (as it is depleted by degradation). This hypothesis is sufficient to explain all effects observed in this part of the paper and is not refuted by any of the data.

11 - Figure S3b - The effect on Nup133 levels is very minor at best.

Rev. 2:

The authors have made important improvements to the manuscript and have done a good job at solidifying data supporting an interaction between Bfa1 and Nup159. There remains considerable doubt, however, as to whether this interaction plays any role in nuclear pore clearance by autophagy. New evidence presented, for example the genetic interaction between NUP159 and BFA1 in the presence of rapamycin, are interesting but without a direct examination of nup degradation, this experiment is open to interpretation. Likewise, although GFP-Atg8 may be less efficiently degraded in nup159-1 cells, the relationship to NPC/nup turnover remains ill defined. As it stands, the definitive Nup159/133-GFP fallout experiments do not demonstrate any clear difference in the degradation of nups in the absence of BFA1 during nitrogen starvation. The authors suggest that this may be due to the presence of Nup159-independent routes to deliver NPCs to the vacuole such as Atg39-dependent nucleophagy and piecemeal microautophagy of the nucleus but neither of these pathways have been demonstrated to degrade whole NPCs and it would have been straightforward to introduce mutants to perturb these pathways. Thus, the strong suggestion is to amend the title to be more circumspect with respect to the role of a Bfa1-Nup159 interaction in NPC phagy.

---

## [Editor Report · Decision Letter 3]

6 May 2023

Dear Dr Monje-Casas,

Thank you for your patience while we considered your revised manuscript entitled "Characterization of a novel interaction of the Nup159 nucleoporin with asymmetrically-localized spindle pole body proteins and its link with autophagy" for publication as a Research Article at PLOS Biology. This revised version of your manuscript has been evaluated by the PLOS Biology editors and the Academic Editor.

Based on our Academic Editor's assessment of your revision, we are likely to accept this manuscript for publication, provided you satisfactorily address the data and other policy-related requests stated below. We would also like you to include the figures for reviewers shown in the rebuttal (A, B and C) as supplementary figures in the manuscript, as they do strengthen your conclusions.

We expect to receive your revised manuscript within two weeks. 

*Published Peer Review History*

*Press*

Sincerely,

Ines

--

Ines Alvarez-Garcia, PhD

Senior Editor

PLOS Biology

Fig. 1D; Fig. 2D; Fig. 3C, E; Fig. 4D, I, J; Fig. 5B, C; Fig. 6B, E, G-I; Fig. S2A-D, F; Fig. S3; Fig. S4D and Fig. S5B-F

We require the original, uncropped and minimally adjusted images supporting all blot and gel results reported in an article's figures or Supporting Information files. We will require these files before a manuscript can be accepted so please prepare and upload them now. Please carefully read our guidelines for how to prepare and upload this data: https://journals.plos.org/plosbiology/s/figures#loc-blot-and-gel-reporting-requirements

---

## [Editor Report · Decision Letter 4]

28 Jun 2023

Dear Dr Monje-Casas,

Thank you for the submission of your revised Research Article entitled "Characterization of a novel interaction of the Nup159 nucleoporin with asymmetrically-localized spindle pole body proteins and its link with autophagy" for publication in PLOS Biology. On behalf of my colleagues and the Academic Editor, Jon Pines, I am delighted to say that we can in principle accept your manuscript for publication, provided you address any remaining formatting and reporting issues. These will be detailed in an email you should receive within 2-3 business days from our colleagues in the journal operations team; no action is required from you until then. Please note that we will not be able to formally accept your manuscript and schedule it for publication until you have completed any requested changes.

PRESS

Sincerely, 

Ines

--

Ines Alvarez-Garcia, PhD

Senior Editor

PLOS Biology
